# Genetic association and causal inference converge on hyperglycaemia as a modifiable factor to improve lung function

William R Reay[1,2], Sahar I El Shair[1], Michael P Geaghan[1,2], Carlos Riveros[2,3], Elizabeth G Holliday[2,3], Mark A McEvoy[2,3†], Stephen Hancock[2,3], Roseanne Peel[2,3], Rodney J Scott[1,2], John R Attia[2,3], Murray J Cairns[1,2]*

[1]School of Biomedical Sciences and Pharmacy, The University of Newcastle, Callaghan, Australia; [2]Hunter Medical Research Institute, Newcastle, Australia; [3]School of Medicine and Public Health, The University of Newcastle, Callaghan, Australia

**\*For correspondence:**
murray.cairns@newcastle.edu.au

**Present address:** [†]La Trobe Rural Health School, College of Science, Health and Engineering, La Trobe University, Bendigo, Australia

**Abstract** Measures of lung function are heritable, and thus, we sought to utilise genetics to propose drug-repurposing candidates that could improve respiratory outcomes. Lung function measures were found to be genetically correlated with seven druggable biochemical traits, with further evidence of a causal relationship between increased fasting glucose and diminished lung function. Moreover, we developed polygenic scores for lung function specifically within pathways with known drug targets and investigated their relationship with pulmonary phenotypes and gene expression in independent cohorts to prioritise individuals who may benefit from particular drug-repurposing opportunities. A transcriptome-wide association study (TWAS) of lung function was then performed which identified several drug–gene interactions with predicted lung function increasing modes of action. Drugs that regulate blood glucose were uncovered through both polygenic scoring and TWAS methodologies. In summary, we provided genetic justification for a number of novel drug-repurposing opportunities that could improve lung function.

## Introduction

Optimal lung (pulmonary) function is vital for the ongoing maintenance of homeostasis, with reduced pulmonary function associated with a marked increase in the risk of mortality (*Vasquez et al., 2017*; *Young et al., 2007*). This is particularly critical due to the considerable number of disorders for which diminished pulmonary function is a clinical hallmark. For instance, chronic obstructive pulmonary disease (COPD), characterised by an irreversible limitation of airflow, is one of the leading causes of death worldwide (*Quaderi and Hurst, 2018*). Pulmonary manifestations are also common amongst disorders not directly classified as respiratory conditions, including diabetes (*Pitocco et al., 2012*; *Walter et al., 2003*), congenital heart disease (*Alonso-Gonzalez et al., 2013*), and inflammatory bowel disease (*Ji et al., 2016*; *Yilmaz et al., 2010*). Bacterial and viral infection, such as *Streptococcus pneumoniae*, *Mycobacterium tuberculosis*, influenza, and coronaviruses, also cause severe declines in respiratory function. In order to better manage the spectrum of respiratory disorders, there is a desperate need for new interventions, including those that can be targeted to an individual's heterogeneous risk factors. While the development pathway for new compounds is difficult, there are likely to be opportunities for precision repurposing of existing drugs to enhance lung function and improve patient outcomes.

**eLife digest** Chronic respiratory disorders like asthma affect around 600 million people worldwide. Although these illnesses are widespread, they can have several different underlying causes, making them difficult to treat. Drugs that work well on one type of respiratory disorder may be completely ineffective on another. Understanding the biological and environmental factors that cause these illnesses will allow them to be treated more effectively by tailoring therapies to each patient.

Reduced lung function is a factor in respiratory disorders and it can have many genetic causes. Studying the genes of patients with reduced lung function can reveal the genes involved, some of which may already be targets of existing drugs for other illnesses. So, could a patient's genetics be used to repurpose existing drugs to treat their respiratory disorders?

Reay et al. combined three methods to link genetics and biological processes to the causes of reduced lung function. The results reveal several factors that could lead to new treatments. In one example, reduced lung function showed a link to genes associated with high blood sugar. As such, treatments used in diabetes might help improve lung function in some patients. Reay et al. also developed a scoring system that could predict the efficacy of a treatment based on a patient's genetics. The study suggests that COVID-19 infection could be affected by blood sugar levels too.

Chronic respiratory disorders are a critical issue worldwide and have proven difficult to treat, but these results suggest a way to identify new therapies and target them to the right patients. The findings also support a connection between lung function and blood sugar levels. This implies that perhaps existing diabetes treatments – including diet and lifestyle changes aimed at reducing or limiting blood sugar – could be repurposed to treat respiratory disorders in some patients. The next step will be to perform clinical trials to test whether these therapies are in fact effective.

Spirometry measures of pulmonary function have been shown to display significant heritability both in twin designs and genome-wide association studies (GWAS) (*Palmer et al., 2001*; *Ingebrigtsen et al., 2011*; *Shrine et al., 2019*). Genomics may reveal clinically relevant insights into the biology underlying lung function, and thus, could be leveraged for drug repurposing. We sought to interrogate the genomic architecture of three spirometry indices to propose drug-repurposing candidates which could be used to improve lung function: forced expiratory volume in 1 s ($FEV_1$), forced vital capacity (FVC), and their ratio ($FEV_1/FVC$). Firstly, we assessed each lung function trait for evidence of genetic correlation with biochemical traits that could be pharmacologically modulated, followed by models to investigate whether there was evidence of causation. The previously developed *pharmagenic enrichment score* (PES) framework was then implemented to identify druggable pathways enriched with lung function-associated variation and calculate pathway-specific polygenic scores (PGS) to prioritise individuals who may benefit from a repurposed compound which interacts with the pathway (*Reay et al., 2020*). A transcriptome-wide association study (TWAS) of $FEV_1$ and FVC was also undertaken to reveal genes which could be targeted by existing drugs that may increase pulmonary function. Finally, we considered the repurposing candidates proposed by these strategies in the context of three respiratory viruses (severe acute respiratory syndrome coronavirus 2 [SARS-CoV2], influenza [H1N1], and human adenovirus [HAdV]), specifically analysing the interactions between viral and human proteins. An overview schematic of this study is detailed in *Figure 1* and *Figure 1—figure supplement 1*.

## Results

### Measures of lung function were genetically correlated with clinically significant metabolites and hormones

We assessed genetic correlation between three pulmonary function measurements ($FEV_1$, FVC, and $FEV_1/FVC$) and 172 GWAS summary statistics of European ancestry using bivariate linkage disequilibrium score regression (LDSC) (*Bulik-Sullivan et al., 2015*; *Zheng et al., 2017*). A number of clinically significant traits displayed significant genetic correlation with $FEV_1$, FVC, and/or $FEV_1/FVC$ after correcting for the number of tests performed ($p < 2.9 \times 10^{-4}$, *Figure 2a*, *Supplementary file 1a–c*).

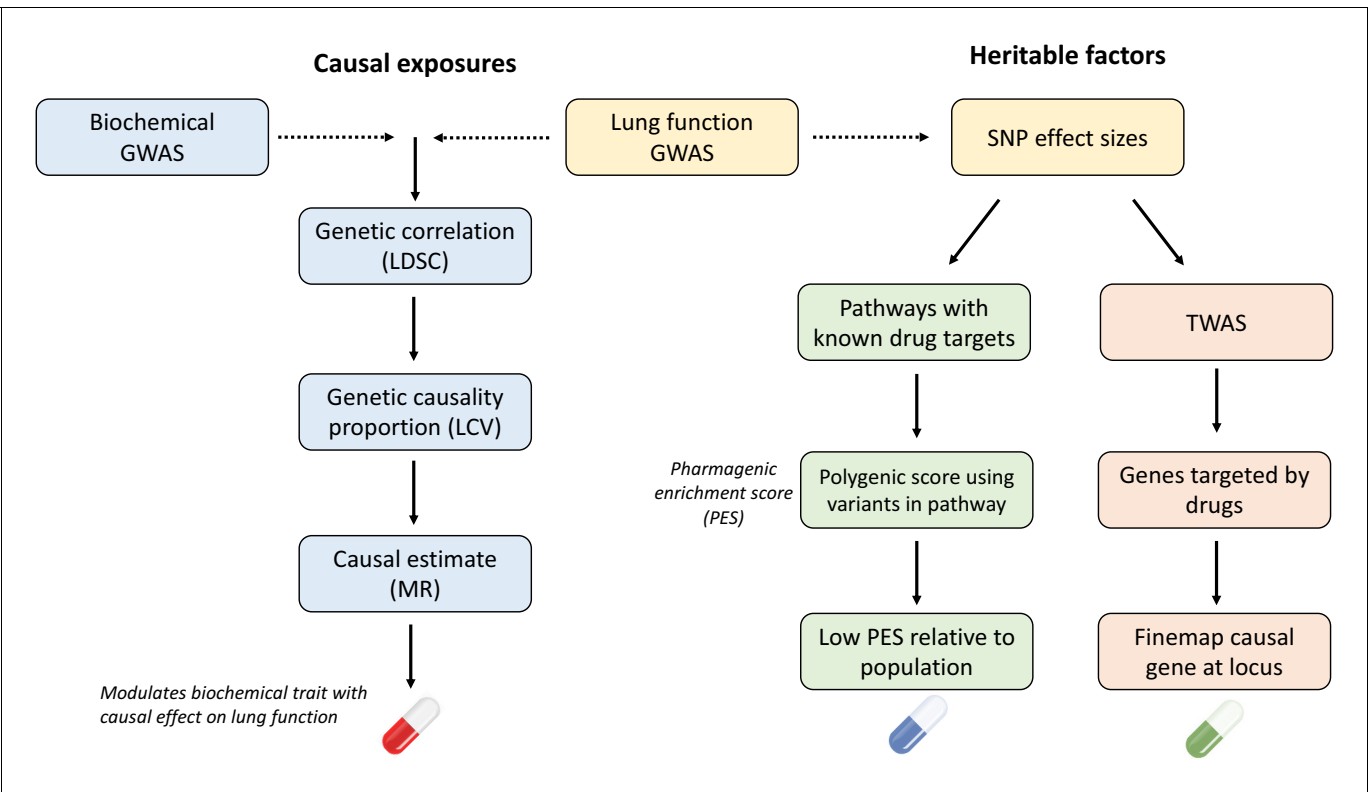

**Figure 1.** Overview of strategies for genetically informed drug repurposing to improve lung function. The left flow chart outlines our workflow for using causal inference to identify drug targets, while the right flow chart shows the workflow for functionally partitioning the heritable component into drug targets. In both cases, we utilise or integrate genome-wide association studies (GWAS) data for lung function (including three spirometry phenotypes: forced expiratory volume in 1 s [$FEV_1$], forced vital capacity [FVC], and their ratio [$FEV_1$/FVC]) and quantitative biochemical traits (e.g. hormones and metabolites) which can be pharmacologically modulated. Using this data, we established genetic correlation between lung function and the biochemical traits using linkage disequilibrium score regression (LDSC). We then constructed a latent causal variable (LCV) model to investigate evidence of causality for significantly correlated biochemical–lung function trait pairs. To further support causal inference between significant pairs, we implemented Mendelian randomisation. Where a causal relationship between a modifiable biochemical trait and lung function is established, we can infer a novel treatment. The right flow chart shows the workflow for utilising heritable components for drug repurposing. Specifically, polygenic scores for lung function were calculated using lung function GWAS single nucleotide polymorphisms (SNPs) within biological pathways that can be targeted by approved drugs, rather than a genome-wide score. Individuals with low genetically predicted lung function by a pharmagenic enrichment score (PES) (low PES) relative to a reference population may benefit from a compound which modulates said pathway. To further support putative genetically predicted targets for drug repositioning a transcriptome-wide association study of lung function was performed. Druggable genes for which genetically predicted expression was correlated with a spirometry measure. Genes with positive genetic covariance between imputed expression and lung function (i.e. increased expression associated with increased lung function) could be modulated by an agonist compound, whilst genes for which decreased predicted expression is associated with improved lung function could be targeted by an antagonist compound.

The online version of this article includes the following figure supplement(s) for figure 1:

**Figure supplement 1.** Diagrammatic overview of strategies for genetically informed drug repurposing to improve lung function.

FVC had the largest number of genetic correlations which surpassed Bonferroni correction (N = 35), followed by $FEV_1$ and $FEV_1$/FVC for which 25 and 8 traits survived multiple testing correction, respectively. The trait most significantly correlated with both $FEV_1$ and FVC was waist circumference – $FEV_1$: $r_g = -0.19$, *SE* = 0.02, p=$5.71 \times 10^{-20}$; FVC: $r_g = -0.24$, *SE* = 0.02, p=$9.54 \times 10^{-33}$ – whilst asthma demonstrated the most significant correlation with $FEV_1$/FVC ($r_g = -0.35$, *SE* = 0.05, p=$3.49 \times 10^{-12}$).

Interestingly, there was evidence of genetic correlation between measures of lung function and circulating levels of both metabolites and hormones. This is notable as these molecules can be pharmacologically modulated, potentially informing novel therapeutic strategies and drug-repurposing opportunities to improve lung function. Significant genetic correlations were observed with four

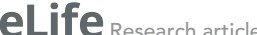

**Figure 2.** Genome-wide investigation of biochemical traits related to lung function. (a) Heatmap of genetic correlations ($r_g$) between three spirometry measures (forced expiratory volume in 1 s [$FEV_1$], forced vital capacity [FVC], and their ratio [$FEV_1$/FVC]) and a number of European ancestry genome-wide association studies. Genetic correlation estimates were plotted if the trait was significantly correlated with at least one of the lung function traits after Bonferroni correction. Hierarchical clustering was applied to the rows and utilised Pearson's correlation distance. (b) Latent causal variable models between correlated biochemical traits (selected by linkage disequilibrium score regression) that are potentially drug targets (metabolite or hormone traits) and each measure of lung function. The posterior mean genetic causality proportion (GCP) is plotted, with the error bars representing the upper and lower limits defined by its posterior mean standard error. A positive GCP estimate significantly different than zero indicates partial genetic causality of the biochemical trait on the spirometry measure.

The online version of this article includes the following figure supplement(s) for figure 2:

**Figure supplement 1.** Investigation of the effect of fasting glucose on lung function using two-sample Mendelian randomisation (MR).

metabolites (fasting glucose, high-density lipoprotein [HDL], triglycerides, and urate) and two hormones (fasting insulin and leptin) for at least one measure of lung function (*Table 1*).

The genetic correlations observed between lung function measures and metabolite/hormone traits may be clinically actionable; however, a significant estimate of genetic correlation does not

**Table 1.** Significant genetic correlations between lung function measures and metabolite and hormone GWAS.

| Lung function trait | Biochemical trait | Genetic correlation ($r_g$)[*] | p-Value |
|---|---|---|---|
| FEV$_1$ | Fasting insulin | −0.23 (0.04) | $6.61 \times 10^{-8}$ |
| | Leptin (BMI unadjusted) | −0.25 (0.05) | $3.74 \times 10^{-7}$ |
| | Leptin (BMI adjusted) | −0.24 (0.05) | $9.13 \times 10^{-7}$ |
| | Urate | −0.12 (0.03) | $9.46 \times 10^{-6}$ |
| | Fasting glucose | −0.13 (0.03) | $1 \times 10^{-4}$ |
| FVC | Fasting insulin | −0.31 (0.04) | $6.98 \times 10^{-14}$ |
| | Leptin (BMI unadjusted) | −0.33 (0.05) | $2.85 \times 10^{-12}$ |
| | Leptin (BMI adjusted) | −0.27 (0.05) | $1.21 \times 10^{-8}$ |
| | HDL cholesterol | 0.14 (0.03) | $9.97 \times 10^{-7}$ |
| | Urate | −0.12 (0.02) | $9.54 \times 10^{-7}$ |
| | Triglycerides | −0.11 (0.03) | $1.53 \times 10^{-5}$ |
| | Fasting glucose | −0.12 (0.03) | $1 \times 10^{-4}$ |
| FEV$_1$/FVC | HDL cholesterol | −0.11 (0.03) | $2 \times 10^{-4}$ |

[*]Genetic correlations which survived multiple testing correction for each lung function trait individually are reported with their respective standard error.

Evidence of a causal relationship between fasting glucose and lung function supports antihyperglycaemic compounds as drug-repurposing candidates.

FEV$_1$: forced expiratory volume in 1 s; FVC: forced vital capacity; FEV$_1$/FVC: ratio of FEV$_1$ to FVC; HDL: high-density lipoprotein; BMI: body mass index; GWAS: genome-wide association studies.

imply causality (*O'Connor and Price, 2018*). In response, we constructed a latent causal variable (LCV) model to estimate mean posterior genetic causality proportion ($\widehat{\text{GCP}}$) for each metabolite or hormone trait and the lung function measure with which it is genetically correlated (*Figure 2b*, *Supplementary file 1d*). The LCV method assumes that a latent variable mediates the genetic correlation between two traits and tests whether this latent variable displays stronger correlation with either of the traits. We used the recommended threshold for partial genetic causality of |GCP| > 0.6 as this has been demonstrated in simulations to appropriately guard against false positives (*O'Connor and Price, 2018*). There was strong evidence of partial genetic causality of fasting glucose on FVC: $GCP$ = 0.77, $SE$ = 0.15, $P_{\text{H0:GCP = 0}}$ = $1.32 \times 10^{-56}$. Importantly, the posterior mean GCP estimate for the relationship between fasting glucose and FVC remained strong (|GCP| > 0.6) using a fasting glucose GWAS additionally adjusted for body mass index (BMI): $GCP$ = 0.63, $SE$ = 0.22, p=$1.67 \times 10^{-56}$. The LCV model constructed between fasting glucose and FEV$_1$ did not surpass the threshold of |GCP| > 0.6 we use to designate partial genetic causality; however, it was directionally consistent with the fasting glucose to FVC estimate and closely approaches this threshold, FEV$_1$: $GCP$ = 0.57, $SE$ = 0.18, $P_{\text{H0:GCP = 0}}$ = $7.18 \times 10^{-12}$. A strong posterior GCP estimate was observed for urate and FVC ($GCP$=0.73), although the relatively low heritability z score as calculated by the LCV framework (z < 7) may lead to a biased estimate. As a result, the relationship between urate and FVC should be treated with caution and further study would be needed to replicate this finding in a urate GWAS with a more precise heritability estimate. The estimate between HDL cholesterol and FEV$_1$/FVC ($GCP$=0.59, $SE$ = 0.26, $P_{\text{H0:GCP = 0}}$ = $4.12 \times 10^{-7}$) was also close to the GCP threshold but we do not denote this as strong evidence of genetic causality given the 0.6 threshold was not exceeded. There was no strong evidence of genetic causality between any of the remaining LDSC-prioritised hormone or metabolite traits and FEV$_1$, FVC, or FEV$_1$/FVC.

As it was the most significant LCV model, the causal effect of fasting glucose on FEV$_1$ and FVC was further investigated utilising a Mendelian randomisation (MR) approach. MR differs from an LCV model as it exploits genome-wide significant variants as genetic instrumental variables (IVs) to calculate a causal estimate of an exposure (fasting glucose) on an outcome (lung function). Given genetic correlation may bias MR due to pleiotropy, we implement MR here as a validation of the LCV results

as it uses different set of statistical parameters and assumptions; however, the estimates derived from MR should be viewed cautiously in light genetic correlation which exists between fasting glucose and lung function (*O'Connor and Price, 2018*). We selected 32 genome-wide significant variants associated with glucose in approximate linkage equilibrium as IVs (p<5×10$^{-8}$, $r^2$ < 0.001) to ensure that variants were both rigorously associated with the exposure and independent from one another. A 1 mmol/L increase in fasting glucose was associated with a −0.088 (95% confidence interval [CI]: −0.17, −0.01) standard deviation decline in FVC using an inverse variance weighted (IVW) estimator with multiplicative random effects. Similarly, elevated fasting glucose was also shown to have a negative effect on FEV$_1$: β$_{IVW}$ = −0.096 (95% CI: −0.18, −0.01). The IVW estimate for fasting glucose was only nominally significant for both FVC and FEV$_1$ (p=0.033 and 0.023, respectively), with relatively wide confidence intervals to approach zero, and thus, the estimate should be treated with appropriate caution. We implemented a number of sensitivity analyses to test the rigour of our causal estimate of the effect of fasting glucose on lung function (*Figure 2—figure supplement 1*, *Supplementary file 1e–g*). Firstly, we obtained an analogous, and statistically significant, causal estimate using the weighted median method (FVC: β$_{Weighted\ median}$ = −0.09 [95% CI: –0.16, –0.04], p=1.87×10$^{-3}$, FEV$_1$: β$_{Weighted\ median}$ = −0.07 [95% CI: –0.13, –0.01], p=0.025). The weighted median method relaxes the assumption that all IVs must be valid, as described elsewhere (*Bowden et al., 2016*). An MR–Egger model was then constructed, which includes a non-zero intercept term which can be used as a measure of unbalanced pleiotropy (*Bowden et al., 2015*). The causal estimate using MR–Egger was in the same direction for FEV$_1$ and FVC; however, it was non-significant (FVC: β$_{MR\ Egger}$ = −0.13 [95% CI: −0.30, 0.04], p=0.148, FEV$_1$: β$_{MR\ Egger}$ = –0.12 [95% CI: −0.30, 0.06], p=0.21). Importantly, the MR–Egger intercept was not significantly different from zero in the FEV$_1$ or FVC model, indicating no evidence of unbalanced pleiotropy. This was supported by a non-significant global test of pleiotropy implemented as part of the MR-Pleiotropy Residual Sum and Outlier (MR PRESSO) framework (*Supplementary file 1f*; *Verbanck et al., 2018*). Furthermore, we evaluated whether there was any evidence of reverse causation, that is, FEV$_1$ and FVC exerting a causal effect on fasting glucose using the MR Steiger directionality test, with our observed direction of causation from glucose to lung function supported.

Finally, we successively recalculated the IVW causal estimate for the effect of fasting glucose on FEV$_1$ and FVC by removing one IV at a time in a 'leave-one-out' analysis (*Supplementary file 1g*; *Burgess et al., 2017*). An analogous causal estimate was derived regardless of which IV was removed; however, there were five IVs (FEV$_1$ model = two outlier single nucleotide polymorphisms (SNPs) , FVC model = four outlier SNPs [two outlier SNPs shared]) for which the estimate was marginally non-significant after exclusion (maximum p=0.11, IVW with multiplicative random effects). We then used a phenome-wide association approach to demonstrate that these five SNPs were (i) annotated to genes with important roles in glycaemic homeostasis and (ii) were almost exclusively associated with glycaemic traits or diabetes (*Supplementary file 1g–l*). As a result, we concluded that these IVs did not likely represent horizontal pleiotropy, which would bias the causal estimate, but instead were biologically salient IVs with large effects (*Supplementary file 1g*).

Whilst smoking status (ever vs. never smoked) was a covariate in the lung function GWAS, we sought to assess whether the relationship between blood glucose and lung function could be driven by residual effects of smoking. There was a significant genetic correlation between the number of cigarettes smoked per day and fasting glucose ($r_g$ = 0.16, SE = 0.043), although this was not observed with the 'ever vs. never smoked' phenotype ($r_g$ = 0.007, SE = 0.039). However, an LCV model constructed for fasting glucose and cigarettes smoked per day did not indicate evidence of genetic causality in contrast to the glucose/lung function models: $GCP$ = −0.47, SE = 0.33, P $_{H0:GCP = 0}$ = 0.25. The MR IVs for glucose were further checked for association with either 'ever vs. never smoked' and 'cigarettes per day', with none of the IVs demonstrating any association with either smoking phenotype at a genome-wide (p<5×10$^{-8}$) or suggestive (p<1×10$^{-5}$) significance threshold (*Supplementary file 1m, n*). Moreover, we investigated the possibility that our results may be impacted by collider bias given the lung function GWAS we utilised was phenotypically covaried for smoking status. We leveraged a smaller UK Biobank GWAS of FEV$_1$ and FVC from the Neale Lab that did not adjust for smoking (N = 272,338). The posterior mean GCP and IVW estimates were in the same direction and relatively analogous for both spirometry measures to that observed using the larger GWAS covaried for smoking status, with no apparent evidence that the negative relationship between glucose and lung function was influenced by smoking as a collider variable. In

summary, these data suggested that there is an effect of fasting glucose on lung function beyond what is directly attributable to a residual impact of smoking.

## Implementation of the *pharmagenic enrichment score* for genetically informed drug repurposing in respiratory distress

We aimed to further expand drug-repurposing opportunities for lung function using the PES approach (*Reay et al., 2020*). Briefly, PES aims to implement genetically informed drug repurposing with PGS calculated using genetic variants specifically within druggable pathways (*Figure 3a*). In the context of this study, individuals with a depleted PES for lung function (lower genetically predicted lung function) mapped to pathways with known drug targets may specifically benefit from drugs which modulate these pathways. Firstly, we performed gene-set association of $FEV_1$ and FVC using a collection of high-quality gene-sets from the molecular signatures database (MSigDB). These sets contain at least one gene which is modulated by an approved pharmacological agent ($N_{Sets}$ = 1030). The $FEV_1$/FVC phenotype is less directly interpretable in this context, given that it is used primarily as a diagnostic tool rather than as a quantitative measure, and thus, we focused on repurposing candidates for $FEV_1$ and FVC individually. Variants were annotated to genes using genomic proximity, with both conservative and liberal upstream and downstream boundary definitions.

Gene-set association using the $FEV_1$ and FVC GWAS was undertaken at each $P_T$ with both conservative and liberal genic boundaries. If a gene-set was significant at multiple $P_T$, the most significantly associated $P_T$ was retained. The conservative genic boundaries only yielded one druggable gene-set enriched with $FEV_1$-associated variants after multiple testing correction ($q < 0.05$): *signalling events mediated by the Hedgehog family* – β = 0.973, *SE* = 0.2, p=9.3×10$^{-7}$, $P_T$ <0.5, $N_{Genes}$ = 22 (*Supplementary file 2a*). There were no gene-sets with known drug targets using conservative genic-boundaries which survived multiple testing correction (false discovery rate (FDR) < 0.05) for association with FVC. Extending the genic boundaries to capture more regulatory variation (liberal boundaries) uncovered more druggable gene-sets (*Supplementary file 2b*). Specifically, there were seven and nine unique gene-sets which survived correction for $FEV_1$ and FVC, respectively ($q < 0.05$, *Table 2*).

It should be noted that there were two pathways related to Hedgehog signalling; however, as these were from different annotation sources and had a different number of genes, we considered them separately. A number of biological processes were encompassed by these prioritised gene-sets, such as cancer (*pathways in cancer, basal cell carcinoma*), transforming growth factor (TGF)-β superfamily signalling (*TGF-β signalling pathway*, *bone morphogenetic protein [BMP] receptor signalling*, *activin receptor-like kinase [ALK] in cardiac myocytes*), and cardiac function (*dilated cardiomyopathy*).

For each candidate PES gene-set, we performed computational drug selection to identify approved compounds predicted to modulate the enriched pathway. Firstly, we investigated U.S. Food and Drug Administration (FDA)-approved pharmacological agents with a statistically significant overrepresentation of target genes in each of these sets ($N_{Overlap}$ ≥3, $q < 0.05$). Drugs which target (i) multiple gene-set members and (ii) more genes than expected by chance were assumed to be particularly relevant for a biological pathway. There were six such gene-sets from the PES candidates which survived multiple testing correction enriched with the targets of an FDA-approved compound (*pathways in cancer*, d*ilated cardiomyopathy*, *class B/2 [secretin family receptors]*, *circadian clock*, *extension of telomeres*, and *extracellular matrix (ECM)/ECM-associated proteins*, *Supplementary file 2c*) – notable drugs included the anti-mineralocorticoid spironolactone, antihyperglycaemic compounds (rosiglitazone, pramlintide), antihypertensives (e.g. verapamil and felodipine), antineoplastic agents (e.g. bexarotene and sunitinib), and nutraceuticals (zinc, vitamin E, and doconexent). Each compound was annotated with its Anatomical Therapeutic Chemical (ATC) classification; the most common first-level ATC code amongst these compounds was antineoplastic and immunomodulating agents (*L*, N = 16), followed by cardiovascular system (*C*, N = 15), and alimentary tract and metabolism (*A*, N = 12; *Figure 3b*). Each of these compounds was subjected to expert curation by a pharmacist in relation to side effects and prior literature evidence as detailed in *Supplementary file 2d* (*Figure 3—figure supplement 1*). Single drug–gene matching was undertaken for remaining PES candidate gene-sets lacking an approved compound with statistically overrepresented target, retaining drug–gene interactions with at least two lines of evidence from Drug–Gene Interaction Database (DGIdb) (*Supplementary file 2e–p*).



**Figure 3.** The *pharmagenic enrichment score* (PES) framework to identify and implement drug-repurposing candidates for lung function. (**a**) Overview of the PES approach, whereby polygenic scores of lung function measures are constructed using variants specifically within druggable pathways. Individuals with a depleted PES, that is, lower genetically predicted spirometry measures using variants in the gene-set, may benefit from a drug which modulates the pathway in question. (**b**) The number of U.S. Food and Drug Administration-approved drugs with overrepresented targets in at least one candidate PES gene-sets per Anatomical Therapeutic Classification (ATC) level 1 code. Each ATC level 1 code is shaded a different colour with its

*Figure 3 continued on next page*

frequency on the x-axis. (c) The phenotypic association between a polygenic score (PGS) of forced vital capacity (FVC) and an FVC PES which was nominally significant (p<0.05) but did not survive multiple testing correction after adjustment for genome-wide PGS. The relationship between the PES/PGS and normalised residual FVC in an independent cohort is plotted, with 95% confidence intervals of the regression trendline indicated by shading. (d) Significant correlations between the expression of genes in a candidate PES and three lung function PES (FVC): *class B/2 secretin family receptors*, *circadian clock*, and *pathways in cancer*. The relationship between PES and gene expression is presented as a volcano plot, where the x-axis is the *t* value (coefficient divided by standard error) and the y-axis is the $-\log_{10}$p-value, with higher points more significant. Genes which are associated after multiple testing correction for the number of genes in the pathway are coloured blue (strict FDR < 0.05) or red (lenient FDR < 0.1). The dotted line denotes an uncorrected nominally significant association (p<0.05).

The online version of this article includes the following figure supplement(s) for figure 3:

**Figure supplement 1.** Schematic for the prioritisation of drug-repurposing candidates.

**Figure supplement 2.** Correlations between genome-wide polygenic scores (PGS) and pharmagenic enrichment score (PES).

In order to test the phenotypic relevance of $FEV_1$ and FVC PES profiles, we utilised an independent genotyped cohort from the Hunter Community Study (HCS, N = 1804). Firstly, we constructed a genome-wide PGS for $FEV_1$ and FVC at six different p-value thresholds (*Supplementary file 3a*). The optimum $FEV_1$ genetic score explained approximately 6.4% of the variance in $FEV_1$ measured in the HCS cohort, whilst the FVC PGS explained approximately 5.7% of variance in FVC. Each of the seven PES profiles were tested for association with $FEV_1$ and/or FVC both with and without adjustment for genome-wide PGS. Four of the PES considered had at least a nominally significant association with their respective spirometry measure (p<0.05, *Table 3*, *Supplementary file 3b*), whilst three survived correction for the number of tests (p<7.14×10$^{-3}$). The variance explained by the significant PES was between 0.4 and 0.7%, with the number of independent SNPs in these scores ranging from 76 to 16,390.

We then constructed a model which was adjusted for genome-wide PGS at the same $P_T$ as the PES and found that only the *class B/2 secretin family receptor* FVC PES remained nominally significant ($\beta$ = 0.047, *SE* = 0.022, p=0.038, *Figure 3c*, *Supplementary file 3c*), although we acknowledge this association does not survive correction for the seven tests performed. This PES did not display

**Table 2.** Gene-sets with known drug targets enriched with lung function-associated common variation after the application of multiple testing correction (FDR < 0.05).

| Phenotype | Gene-set | Lowest p* | Genic boundaries |
|---|---|---|---|
| FVC | Hedgehog signalling pathway (KEGG) | $6.66 \times 10^{-9}$ | Liberal |
| | BMP receptor signalling | $4.08 \times 10^{-7}$ | Liberal |
| $FEV_1$ | Signalling events mediated by the Hedgehog family | $9.30 \times 10^{-7}$ | Conservative |
| | Hedgehog signalling pathway (KEGG) | $3.45 \times 10^{-6}$ | Liberal |
| FVC | ALK in cardiac myocytes | $4.57 \times 10^{-6}$ | Liberal |
| | Pathways in cancer | $5.43 \times 10^{-6}$ | Liberal |
| $FEV_1$ | Basal cell carcinoma | $8.86 \times 10^{-6}$ | Liberal |
| FVC | TGF-β signalling pathway | $1.21 \times 10^{-5}$ | Liberal |
| | Circadian clock | $3.00 \times 10^{-5}$ | Liberal |
| | Class B/2 (secretin family receptors) | $8.08 \times 10^{-5}$ | Liberal |
| $FEV_1$ | TGF-β signalling pathway | $8.15 \times 10^{-5}$ | Liberal |
| | Extension of telomeres | $8.59 \times 10^{-5}$ | Liberal |
| | Pathways in cancer | $8.94 \times 10^{-5}$ | Liberal |
| | Dilated cardiomyopathy | $9.54 \times 10^{-5}$ | Liberal |
| FVC | ECM/ECM-associated proteins | $2.28 \times 10^{-4}$ | Liberal |

*The lowest p is the most significant gene-set association p-value across all the p-value thresholds ($P_T$) and genic boundary configurations tested.

ECM: extracellular matrix; FVC: forced vital capacity; $FEV_1$: forced expiratory volume in 1 s: TGF-transforming growth factor; ALK: activin receptor-like kinase; BMP: bone morphogenetic protein.

**Table 3.** The association between lung function PES and spirometry measures in the Hunter Community Study cohort.

| Phenotype | PES | Z value | p | PES $R^2$ | $N_{SNP}$ |
|---|---|---|---|---|---|
| FEV$_1$ | Dilated cardiomyopathy | 0.15 | 0.889 | $1.3 \times 10^{-5}$ | 2404 |
| | Extension of telomeres | −0.18 | 0.861 | $1.7 \times 10^{-5}$ | 44 |
| | Pathways in cancer | 2.98 | 0.003 | 0.005 | 6214 |
| FVC | Circadian clock | 2.14 | 0.033 | 0.003 | 230 |
| | Class B/2 secretin family receptors | 3.14 | 0.002 | 0.005 | 76 |
| | Extracellular matrix proteins | 3.50 | $5 \times 10^{-4}$ | 0.007 | 16,390 |
| | Pathways in cancer | 2.64 | 0.008 | 0.004 | 6212 |

PES: pharmagenic enrichment score;
FVC: forced vital capacity;
FEV$_1$: forced expiratory volume in 1 s.

The $Z$ value is the PES model coefficient divided by its standard error. The variance explained ($R^2$) was the null model $R^2$ subtracted from the full model with the PES as a predictor. The number of independent SNPs used to calculate the PES in this cohort is reported in the $N_{SNP}$ column. The reported results are from models unadjusted for genome-wide PGS.

any association with smoking status in this cohort ($\beta = -0.014$, *SE* = 0.047, p=0.758), whilst the signal remained nominally significant upon removing HCS participants with self-reported respiratory illness (N = 1433, $\beta = 0.052$, *SE* = 0.025, p=0.042). Furthermore, there was a significant depletion of FVC within the 10th percentile (low genetically predicted FVC) of the *class B/2 secretin receptor family* FVC PES in the HCS cohort, with the odds of being in the lowest decile decreasing by around 20% per standard deviation increase in FVC (OR = 0.80 [95% CI: 0.68, 0.93], p=4.7×10$^{-3}$). All of the PES tested demonstrated small albeit significant correlations with genome-wide PGS at the same $p_T$ in the HCS cohort, with the exception of the *extracellular matrix* PES for which the correlation was relatively large (r = 0.33, *Figure 3—figure supplement 2*). The higher correlation in this gene-set was probably due to the large number of genes involved (>1000). Interestingly, there was still a number of individuals with high genetically predicted lung function using a genome-wide PGS (90th percentile of HCS cohort) but low genetically predicted lung function using one of the PES (10th percentile). Specifically, 12.17% and 12.05% of the HCS participants in the 90th percentile PGS for FVC and FEV$_1$ respectively had a depleted PES (10th percentile, low predicted lung function by PES). Taken together, this suggests that pathway-based PGS provide distinct biological insights for some individuals with otherwise high genetic load of lung function increasing alleles, although the association between the *class B/2 secretin family receptor* PES and FVC after covariation for PGS did not survive multiple testing correction, and thus, these data require replication.

The correlation between the expression of genes within each pathway encompassed by the PES and the PES profiles themselves could provide further support for their biological impact. We investigated the association between lung function PES and gene expression using RNA sequencing (RNA-seq) on transformed lymphoblastoid cell lines (LCL) from 357 European individuals for which phase 3 whole-genome sequencing data was available from the 1000 genomes project (*Figure 3d*, *Supplementary file 3d–j*; *Lappalainen et al., 2013*). We identified a significant association between the FVC PES *class B/2 [secretin family receptors]* and the expression of *WNT3* using a strict FDR threshold q < 0.05 (*t* = −3.53, p=4.71×10$^{-4}$, q = 0.028); a more lenient FDR cut-off (q < 0.1) yielded two more significant PES–gene expression correlations: FVC *circadian clock* PES and *PPARA*: *t* = −3.23, p=1.37×10$^{-3}$, q = 0.07; FVC *pathways in cancer* and *HSP90AB1*: *t* = 3.72, p=2.33×10$^{-4}$, q = 0.066. Expression of *WNT3* and *PPARA* was not associated with genome-wide PGS at the same p-value threshold (p=0.63 and 0.29), whilst the PGS exhibited a weaker, nominal relationship with *HSP90AB1* (p=0.04). The remaining four PES tested (FEV$_1$ or FVC) all demonstrated at least one nominal, uncorrected association (p<0.05). The observed effects of PES on gene expression at the population level were subtle; this is not surprising as each PES profile will encompass heterogenous variants for each individual, and thus, impacts on gene expression may be greater within specific genomic contexts.

# Transcriptome-wide association identifies putative targets for pharmacological modulation of lung function

We performed a TWAS of the three lung function measures using SNP weights from lung and blood tissue. TWAS leverages models of genetically regulated expression to test for a correlation between predicted expression and a phenotype (*Gusev et al., 2016*). Models of imputed expression derived from *cis*-eQTLs are generated from genes for which expression displays significant *cis*-heritability, that is, a significant genetic contribution to expression variance. We aimed to identify genes for which increased or decreased expression was associated with increased lung function and had approved compounds available which could improve lung function based on their mechanism of action (*Figure 4a*). For instance, if increased expression of a gene was associated with improved lung function, then an agonist of that gene may be clinically useful or vice versa in the case of decreased expression. Using a Bonferroni threshold for the number of genes tested in lung and blood individually, we identified a number of transcriptome-wide significant genes as follows – $FEV_1$: $N_{Genes\ [Lung]}$=232, $N_{Genes\ [Whole\ blood]}$=201; FVC: $N_{Genes\ [Lung]}$=222, $N_{Genes\ [Whole\ blood]}$=167 (*Supplementary file 3k–n*, *Figure 4b*). The number of significant genes remained very similar using

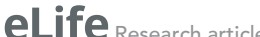

**Figure 4.** The application of transcriptome-wide association to identify drug-repurposing candidates for lung function. (**a**) Schematic outlining the use of transcriptome-wide association study (TWAS) to reveal clinically actionable drug–gene interactions. Druggable genes with lung function-associated imputed expression can be finemapped to prioritise a credible set of a causal genes at the TWAS locus, that is, a high posterior inclusion probability (PIP). We seek to identify drugs with a mode of action which match the TWAS *Z* value, that is, compounds which may increase lung function. (**b, c**) Miami plots of a TWAS of forced expiratory volume in 1 s (left) and forced vital capacity (right) using whole blood (**b**) and lung (**c**) SNP weights. TWAS *Z* > 0 denotes a gene for which increased predicted expression is associated with increased lung function and vice versa. The highlighted genes survived multiple testing correction for the number of genes tested. (**d**) Probabilistic finemapping of the *PYGB* TWAS locus. The points denoting each gene are sized and coloured by their PIP for causality, with higher PIP denoted by larger, darker points as represented on the scale. The correlation plot below each region represents the covariance of predicted expression between gene.

a more conservative threshold for Bonferroni correction that accounted for all genes in both tissues ($p<3.6\times10^{-8}$), which is conservative due to correlation between imputed models (*Supplementary file 3k–n*). Transcriptome-wide associated genes were only retained if they were not also associated with a smoking phenotype to minimise residual smoking-related confounding. Specifically, we tested whether predicted expression of the genes which survived correction in the $FEV_1$ or FVC TWAS was associated with smoking behaviour ('ever vs. never smoked' and 'cigarettes per day') in a TWAS using SNP weights from lung, blood, and two brain regions implicated in nicotine addiction (dorsolateral prefrontal cortex and nucleus accumbens; *Supplementary file 3m–v*; *Goldstein and Volkow, 2011*; *Scofield et al., 2016*). We searched each of these significant genes in the DGIdb v3.0.2 to ascertain compounds which may improve lung function based on the direction of effect from the TWAS analyses. In accordance with the PES analyses, $FEV_1/FVC$ was not directly considered and we focused on $FEV_1$ and/or FVC-associated genes which could be pharmacologically modulated (*Supplementary file 3w*).

Four candidate genes were identified satisfying tier one criteria: *PPARD*, *ADORA2B*, *KCNJ1*, and *AMT*. For instance, decreased expression of potassium channel gene *KCNJ1* was associated with FVC ($Z_{TWAS} = -4.60$), and this channel can be inhibited by approved compounds such as the antidiabetic drug glimepiride. There were an additional seven genes with tier 2 investigational targets: *PYGB*, *PIK3C2B*, *LINGO1*, *APH1A*, *OPRL1*, *MST1R*, and *ACVR2B*. Probabilistic finemapping of these transcriptome-wide significant regions using a multi-tissue reference panel was then performed to prioritise whether these genes are likely causal at that locus. A credible set with 90% probability of containing the causal gene was computed for each locus utilising the marginal posterior inclusion probability (PIP) calculated from the observed TWAS statistics. We did not proceed with finemapping the *PPARD* locus due to its proximity to the defined boundaries of the major histocompatibility complex (MHC) region. Two $FEV_1$-associated genes with tier 1 and/or tier 2 drug interactions, *AMT* and *PYGB*, were included in the credible set with a PIP >0.9 or nearing that threshold. Tetrahydrofolate is a co-factor for *AMT* ($Z_{TWAS} = 5.96$, $\mathrm{PIP} = 0.893$, whole blood SNP weights), which has been previously implicated as having a beneficial effect on lung function. *PYGB* ($Z_{TWAS} = -6.98$, $PIP = 0.999$, lung SNP weights) encodes a protein involved in glycogenolysis and can be putatively inhibited by the new exploratory treatment for respiratory failure, sivelestat (*Figure 4c*). We acknowledge that the interaction between *PYGB* and sivelestat was derived from two public databases curated by DGIdb v.3.0.2 (*Supplementary file 3w*), and appropriate caution should be exercised in interpreting this relationship given that *PYGB* is not the primary target of sivelestat. Interestingly, there is evidence of a high-confidence biological interaction between *PYGB* and the gene that encodes the principal target of sivelestat via the STRING database (*ELA2*, neutrophil elastase).

In addition, we tested a more conservative Bernoulli prior for each causal indicator ($p=1\times10^{-5}$) but this only had a negligible effect on the PIP for either *AMT* ($PIP = 0.87$) or *PYGB* ($PIP = 0.994$). Whilst there is a plausible role for *AMT* in respiratory biology (aminomethyltransferase, involved in glycine cleavage), it should be noted that decreased predicted expression of *AMT* also trended towards the Bonferroni threshold for a significant association with smoking status ($Z_{TWAS} = -4.33$, $p=1.46\times10^{-5}$), although this was weaker for the cigarettes per day phenotype ($Z_{TWAS} = -2.97$, $p=2.94\times10^{-3}$). As a result, the association of this region with $FEV_1$ should be treated cautiously until its biological relevance can be clarified to ensure that this signal is not driven by a residual effect of smoking.

## Host–viral interactomes suggested proposed pulmonary drug-repurposing candidates may be significant for respiratory virus infection

Respiratory viruses are an important contributor to acute, and potentially fatal, declines in lung function. We sought to investigate whether our proposed drug-repurposing candidates for lung function may also exhibit antiviral properties against these pathogens. The host–virus interactome was analysed for three respiratory viruses to perform computational drug repurposing – SARS-CoV2, H1N1, and the HAdV family (*Supplementary file 4a–c*; *Gordon et al., 2020*; *Watanabe et al., 2014*; *Martinez-Martin et al., 2016*). Specifically, human proteins which are predicted to interact with virally expressed proteins ('prey proteins') were investigated to identify those which could be inhibited by existing drugs to potentially disrupt the progression of infection. Approved inhibitors or antagonists of proteins in each respective host–virus interactome were sourced using DGIdb and compared to

our candidate compounds for lung function from the PES approach. Furthermore, we investigated the reported drug-label side-effect frequencies of each of these overlapping pharmacological agents and retained only candidates with no commonly reported (>1% frequency) respiratory adverse effects. There were three inhibitors of human proteins with evidence of interaction with a viral protein that also targeted a gene which was a member of a PES candidate gene-set. Vorinostat (*HDAC2* inhibitor) and aminocaproic acid (*PLAT* inhibitor) both inhibited a SARS-CoV2 'prey protein' and targeted a gene within the *pathways in cancer* and *extracellular matrix (ECM)/ECM-associated proteins* PES pathways, respectively. Similarly, ruxolitinib inhibits the influenza prey protein *JAK1*, a part of the *pathways in cancer* gene-set. We caution that these pathways are quite broad in the biology that they encompass, and, as a result, the relevance of these drug–gene interactions to the pathways of interest warrants further study.

We demonstrated using multiple lines of evidence a putative relationship between increased fasting blood glucose and lung function; therefore, we investigated whether any of the host–viral interactome members were enriched within biological pathways involved in glycaemic homeostasis. Interestingly, there was an overrepresentation of SARS-CoV2 'prey proteins' amongst four gene-sets related to glucose metabolism, along with insulin and glucagon signalling pathways (*Table 4*). Fourteen SARS-CoV2 'prey proteins' were members of at least one of these gene-sets, with a greater number of interactions amongst these genes than expected by chance (p=$4.42 \times 10^{-12}$, *Supplementary file 4d*). We outline evidence for the potential role of these viral prey genes in glycaemic homeostasis in *Supplementary file 4d*. These data support emerging evidence that SARS-CoV2-infected patients with hyperglycaemia are at higher risk of morbidity and mortality (*Kumar et al., 2020*).

None of the glycaemic 'prey proteins' were direct target of antidiabetic compounds; however, 57% of these proteins had a high-confidence protein–protein interaction with antidiabetic target gene (*Supplementary file 4e–f*). For instance, *GNB1* putatively binds with a SARS-CoV2 non-structural proteins (Nsp7) that forms the part of the replicase/transcriptase complex, whilst this protein also demonstrated evidence of interacting with 15 proteins modulated by an antidiabetic compound, such as *GLP1R*, which is the primary target of GLP-1 analogues, including exenatide. Pharmacological interventions which seek to control blood glucose may have positive implications both in terms of improving baseline lung function and reducing the risk of adverse consequences after SARS-CoV2 exposure.

## Discussion

We revealed candidate drug-repurposing opportunities to potentially improve pulmonary function and provide the means for aligning their application in individuals that carry a high relative burden of variants associated with their function. Through this process we identify glycaemic interventions in particular as being potentially beneficial in the context of respiratory infection. Our study suggests a causal relationship between blood glucose and lung function using a genome-wide (LCV) and IV (MR) approach, whilst downregulation of the glycogen phosphorylase *PYGB* was also associated with FEV$_1$ after probabilistic finemapping of TWAS loci. These data support previous literature suggesting that declines in pulmonary function are overrepresented amongst individuals with diabetes and correlates with poor glycaemic control (*Walter et al., 2003*; *Davis et al., 2004*; *Gutiérrez-Carrasquilla et al., 2019*; *van den Borst et al., 2010*); a phenomenon which has also been reported in non-diabetics (*McKeever et al., 2005*; *Barrett-Connor and Frette, 1996*). There are a number of

**Table 4.** Overrepresentation of proteins which interact with viral severe acute respiratory syndrome coronavirus 2-expressed proteins within glycaemic-related pathways.

| Glycaemic gene-set | p-Value |
| --- | --- |
| Glucagon-like peptide-1 regulates insulin secretion | $7.02 \times 10^{-4}$ |
| Glucagon signalling in metabolic regulation | $2.33 \times 10^{-4}$ |
| Glucose metabolism | $2.69 \times 10^{-5}$ |
| Regulation of insulin secretion | $2.13 \times 10^{-3}$ |

pathophysiological mechanisms postulated to underlie this relationship, including fibrosis mediated by hyperglycaemia-accelerated epithelial-to-mesenchymal transition (*Talakatta et al., 2018*) and aberrant inflammatory responses to dysglycaemia (*Mohanty et al., 2000*; *Sun et al., 2014*). Importantly, our data extends on these previous observational studies to provide novel evidence for a causal relationship. Respiratory sequalae after infection may also be significantly affected by dysregulation of glycaemic control. Acute hyperglycaemia is associated with a significant increase in morbidity and mortality amongst non-diabetic community-acquired pneumonia (CAP) patients, which further supports its utility as a treatment target (*Lepper et al., 2012*; *Jensen et al., 2017*; *Kornum et al., 2007*; *McAlister et al., 2005*). Notably, even patients with mild hyperglycaemia (serum glucose 6–10.99 mmol/L) have a purported elevated risk of death at 90 days following CAP diagnosis (*Lepper et al., 2012*), whilst the association between type 2 diabetes and poor pneumonia outcomes appears to be driven by glycaemic control (*McAlister et al., 2005*). Inflammation is likely to be an important component of glycaemic-influenced adverse effects; for instance, the intracellular carbohydrate $O$-linked β-$N$-acetylglucosamine has been recently linked to influenza-associated cytokine storms (*Wang et al., 2020*). Future work should focus on the relevance of glycaemic biology to specific respiratory illnesses like asthma and COPD. Our findings supported the relevance of glycaemia to respiratory infection through demonstrating that proteins which putatively interact with the SARS-CoV2 virus were overrepresented in glycaemic pathways. Whilst the viral prey proteins we identified as members of glycaemic pathways were not the direct targets of antihyperglycaemic agents, some interact with these compounds, although the biological saliency of these interactions warrants future investigation. The presence of a viral–prey protein interaction also does not necessarily support its essentiality in the viral life cycle, and further data are needed to support this. Furthermore, the viral prey proteins overrepresented in the glycaemic pathways were mostly genes such as nucleoporins and cAMP-dependent protein kinases which have pleiotropic regulatory roles spanning a number of biological systems. It would also be interesting to further explore the relationship between the genetic architecture of fasting glucose and expression of these SARS-CoV2 prey proteins. These data taken together support the utility of managing blood glucose in the clinical improvement of respiratory outcomes.

Targeted drug application and repurposing is by its very nature confounded by biological heterogeneity amongst individuals. This is likely particularly true in the case of complex traits as their polygenic genetic architecture provides the substrate for each individual to display a unique profile of trait-associated variation. In the second stream of this study, we stratified the polygenic architecture of lung function into a series of druggable pathways to provide a framework for pathway-specific genetic scores we designate the PES. We suggest that leveraging inter-individual genetic heterogeneity in this way will improve the precision application of novel drug repurposing. A number of interesting drug-repositioning candidates had overrepresented targets amongst the candidate PES gene-sets. For example, magnesium sulfate had enriched targets in the *dilated cardiomyopathy* PES and has previously shown promise as a repurposing candidate to improve pulmonary function in asthma (*Okayama, 1987*; *Hossein et al., 2016*). Using an independent cohort, several PES profiles tested explained a small, but significant, percentage of variance in FEV$_1$ and/or FVC. The *class B/2 secretin family receptors* score for FVC was noteworthy given that it remained nominally significant after an adjustment for genome-wide PGS. However, this did not survive multiple testing correction, and thus, further replication is needed to confirm this signal. Interestingly, this gene-set features a number of proteins involved with glycaemic homeostasis, including antidiabetic drug targets glucagon-like peptide receptor 1 (*GLP1R*) and amylin receptors (*RAMP1*, *RAMP2*, and *RAMP3*). While all of the PES demonstrated significant correlation with genome-wide PGS, in the majority of cases it was small ($r < 0.2$), suggesting that most of these functionally relevant foci of genomic risk in lung function GWASs were relatively independent of the total PGS. Importantly, we still identified individuals with high genetically predicted lung function using a genome-wide PGS but observed low predicted lung function with a pathway-specific PES. This was supported by the observed correlation between the PES and related mRNA expression which was distinct from a genome-wide PGS. Collectively, these data are consistent with the hypothesis that important treatment-related biology could be captured at a pathway level for individuals with or at risk of respiratory illness. The specific data from this study require future replication and validation in independent cohorts in order to provide greater support to our observed relationships between PES and spirometry measures. In addition, further study is warranted to dissect the signals encompassed by pathway-specific PGS,

particularly in light of what would be observed amongst other pathways without drug targets or in gene-sets associated with other related traits.

Taken together, our approach provides template for genetically informed precision drug repositioning to improve lung function. The clinical implementation in its most basic form would involve common variant genotyping using a commercial SNP array followed by imputation and lung function PES-based stratification of treatment options. This would be combined with other biochemical exposure measures, such as fasting glucose, that are causal risk factors and have approved treatments. To illustrate the clinical implementation of our strategy, we generated a schematic representation of individual heterogeneity in biochemical and genetic components of risk in lung function and related them to candidates for precision drug repositioning (*Figure 5*). We envisage that our approach to variant and exposure risk stratification can be applied more broadly to identify and implement precision drug repositioning in a range of complex traits.

Whilst there are some potential confounds in the use of GWAS data for causal inference via both LCV models and MR, such as measurement error, population stratification, and horizontal pleiotropy, we are confident that the relationship between glycaemia and lung function presented in this study is robust given the multiple lines of support. Replicated, well-powered randomised controlled trials, however, are needed to fully resolve the clinical benefit of repurposing antihyperglycaemic compounds to improve lung function and in the context of viral infection. We also acknowledge that the direction of suitable pharmacological intervention is not inherently clear, such that an agonist or antagonist of genes within a pathway implicated by the PES approach is an important consideration (*Reay et al., 2020*). Careful curation of the proposed repurposing candidates will therefore be critical, particularly in the context of pulmonary traits where a variety of currently approved compounds have adverse respiratory effects. We suggest that TWAS could be utilised to help overcome these issues by identifying druggable genes which are members of candidate PES gene-sets for which a clinically beneficial impact on expression can be predicted. These candidate genes derived from TWAS could in future be explored further using well-powered cohorts with genetic and transcriptomic that recorded spirometry measures. Interestingly, we also saw some evidence of cross-talk between heritable risk at genes associated with lung function and fasting glucose, with the downregulation of the glycogen phosphorylase *PYGB* (associated with $FEV_1$) observed through the probabilistic finemapping of TWAS loci.

This study demonstrated a variety of methods for which genomic data could be utilised to propose drug-repurposing candidates, ranging from approaches which exploit genome-wide variant effects to the identification of candidate clinically significant drug–gene interactions. Lung function is a particularly relevant phenotype to study in this context as its aetiology is influenced by a variety of complex biological factors, and it is a significant contributor to global morbidity and mortality. Genetics-informed approaches will likely be increasingly useful to target novel respiratory interventions and reposition existing compounds. In future, genetics-based methods could be integrated with other clinical information to further enhance precision drug repurposing, whilst further consideration could be given to experimental compounds to enhance the number of repurposing opportunities. Our data strongly supported the efficacy of antihyperglycaemic compounds as repurposing candidates which could act as the impetus for further clinical investigation via randomised controlled trials.

## Materials and methods

### Lung function GWAS

We obtained GWAS summary statistics for $FEV_1$, FVC, and their ratio from a meta-analysis of the UK Biobank sample with the SpiroMeta consortium cohorts as outlined extensively elsewhere (N = 400,102) (*Shrine et al., 2019*). Phenotypes were adjusted for age, $age^2$, sex, height, smoking status (ever vs. never smoked), and genotyping array before the residuals were subjected to rank inverse-normal transformation. This GWAS was performed using European ancestry individuals.

### Genetic correlation

Bivariate LDSC regression was performed between each lung function trait and a variety of GWAS as implemented by LDhub v1.9.3 (*Zheng et al., 2017*). Lung function summary statistics were



**Figure 5.** Schematic representation of drug repositioning and precision implementation in lung function deficits directed by causal enrichment of environmental and genetic risk factors. Each row represents a simulated individual with a heterogeneous presentation of risk factors related to lung function. Case 1 (top row) represents an individual with good lung function (pink lung tissue) and genomic and environmental components consistent with healthy lung function (grey to red nodes). These have a neutral to positive influence on lung function represented by the grey and red edges (arrow), respectively. Case 2 has high fasting glucose and neutral (grey) loading of genetic variants (pharmagenic enrichment score [PES]) associated with lung function pathways. After treatment with antihyperglycaemic agents, or some other intervention to lower blood glucose, lung function is improved (red edge) sufficiently for therapeutic effect, represented by pink lungs. Case 3 has enrichment of genetic variants (PES) associated with poorer lung function in the *class b2 secretin* pathway. To improve lung

*Figure 5 continued on next page*

*Figure 5 continued*

function, they are treated with drugs, such as pramlintide (which targets *RAMP1*, *RAMP2*, and *RAMP3*) and exenatide (*GLP1R* agonist), which works by modulating genes in the *class b2 secretin* pathway to ameliorate the enrichment of poor lung function variants in that pathway. The broken edge between fasting glucose and the *class b2 secretin* pathway represents the probable connection or shared genes between these nodes as receptors in this pathway are involved in glycaemic regulation. Case 4 also presents with poor lung function (blue lung tissue) and enrichment of poor lung function-associated variants in the *circadian clock* pathway (blue node). This individual's lung function was then treated by compounds, such as doconexent, which act on the *circadian clock* pathway. This schematic is only representative of many thousands of treatment scenarios potentially informed by this treatment decision tool, which could be applied to any phenotype with large genome-wide association studies available.

cleaned ('munged') prior to LDSC using munge_sumstats.py and merged with common HapMap3 SNPs excluding the MHC region due to its LD complexity, as is usual practice (*Bulik-Sullivan et al., 2015*). We retained estimates of genetic correlation ($r_g$) for GWAS (N = 172) with European ancestry and a heritability *Z* value >4, as calculated by LDhub. When a phenotype had multiple GWAS, the GWAS with largest sample size was retained. The Bonferroni method was utilised for multiple testing correction with the significance threshold set as $p < 2.9 \times 10^{-4}$ ($\alpha$ = 0.05/172). A heatmap was constructed using the ComplexHeatmap package (*Gu et al., 2016*).

## LCV models

LCV models were constructed between each measure of lung function which displayed a significant genetic correlation with a hormone or metabolite trait. The RunLCV.R and MomentFunctions.R scripts were leveraged to perform these analyses (https://github.com/lukejoconnor/LCV). The LCV framework assumes that a latent variable, *L*, mediates the genetic correlation between two traits (trait 1, trait 2) and uses the mixed fourth moments of the bivariate effect size distribution to estimate the mean posterior GCP as described in detail by *O'Connor and Price, 2018*. The GCP estimate quantifies the magnitude of genetic causality between the two traits. GCP values range from −1 to 1 (full genetic causality); within these limits, positive values indicate greater partial genetic causality of trait 1 on 2, and vice versa for negative values. All traits were munged prior to LCV analyses, with only HapMap3 SNPs (minor allele frequency [MAF] >0.05) outside the MHC region retained in accordance with the LDSC analyses. We utilised the baseline 1000 genomes phase 3 LD scores for HapMap3 SNPs (MHC excluded). A two-sided *t*-test was used to assess whether the estimated GCP was significantly different from zero.

## Mendelian randomisation

We investigated the causal effect of fasting glucose on both FEV$_1$ and FVC using two-sample MR. MR is underpinned by the use of genetic variants as IVs, with the random inheritance of these IVs as per Mendel's laws facilitating the use of IVs to perform causal inference between an exposure and outcome, providing a series of assumptions are met (*Burgess et al., 2017*). We defined IVs as independent variants which are associated with fasting glucose using the traditional GWAS genome-wide significance threshold ($p < 5 \times 10^{-8}$, $r^2 < 0.001$, palindromic SNPs removed). A different GWAS of fasting glucose was utilised for MR than for LDSC and LCV. Scott et al. performed a replication of ~66,000 Illumina CardioMetabochip variants following the Manning et al. GWAS for which more complete summary statistics were available, and thus, the latter was included in the LDhub catalogue instead of the former (*Manning et al., 2012*; *Scott et al., 2012*). We required only genome-wide significant SNPs for MR; therefore, the Scott et al. CardioMetabochip replication was more suitable as this was a larger sample size than the Manning et al. GWAS. Fasting glucose data for GWAS were obtained from either plasma or whole blood of non-diabetic individuals of European ancestry and corrected to plasma levels (N = 133,310, unit of effect = mmol/L) (*Scott et al., 2012*). Our primary MR model was an IVW effect model with multiplicative random effects (*Burgess et al., 2013*). Further, we implemented a weighted median model which takes the median of the ratio estimates (as opposed to the mean in the IVW model), such that upweighting was applied to ratio estimates with greater precision (*Bowden et al., 2016*). An MR–Egger model was then constructed; an adaption of Egger regression wherein the exposure effect is regressed against the outcome with an intercept

term added to represent the average pleiotropic effect (*Bowden et al., 2015*). In addition, we examined evidence of reverse causality by using the MR Steiger directionality test (*Hemani et al., 2017*).

We also tested whether the Egger intercept is significantly different from zero as a measure of unbalanced pleiotropy. In addition, heterogeneity amongst the IV ratio estimates was quantified using Cochran's $Q$ statistic, given that horizontal pleiotropy may be one explanation for significant heterogeneity. A global pleiotropy test was also implemented via the MR PRESSO framework (*Verbanck et al., 2018*). Leave-one-out analyses were then performed to assess whether causal estimates are biased by a single IV, which may indicate the presence of outliers, and the sensitivity of the estimate to said outliers. However, outliers may not necessarily be evidence of horizontal pleiotropy. We performed a phenome-wide association study for each of these 'outlier' SNPs using summary data collated by GWAS atlas v20191115 to assess evidence of horizontal pleiotropy, that is, acting through non-glycaemic pathways to influence lung function (*Watanabe et al., 2019*). All MR analyses were performed in R version 3.6.0 using the TwoSampleMR v0.4.25 and MRPRESSO v1.0 packages.

## Investigating residual confounding from smoking on the relationship between fasting glucose and lung function

We investigated whether a residual effect of smoking could confound the link between glucose and lung function. Firstly, we selected two well-powered GWAS of smoking behaviours: ever vs. never smoked (N = 385,013) (*Watanabe et al., 2019*) and cigarettes smoked per day (N = 263,954) (*Liu et al., 2019*). Genetic correlation between these two smoking phenotypes and fasting glucose was estimated as described above, followed by the construction of an LCV model. The MR IVs utilised for fasting glucose were also checked for association with each smoking GWAS. We also probed whether there could be an effect of collider bias in the event smoking does indeed exert an effect on fasting glucose. To this end the LCV and MR analyses was repeated for fasting glucose using smaller UK Biobank GWAS of $FEV_1$ and FVC from the Neale Lab as it was not adjusted for smoking (N = 272,338, http://www.nealelab.is/uk-biobank).

## Generation of PES candidate gene-sets

We implemented gene-set association using MAGMA method (MAGMA v1.06b), with some customisations to the framework to identify candidate PES gene-sets (*Reay et al., 2020*; *de Leeuw et al., 2015*). These gene-sets became the basis to calculate pathway-specific polygenic scores (PES). MAGMA aggregates SNP-wise p-values for trait association into a gene-based p-value and, thereafter, tests whether a set of genes is more strongly associated with the phenotype than all other genes. Gene-based test statistics were calculated analogous to Brown's method, which is applicable to dependent p-values with known covariance (as common SNPs display through the phenomenon of linkage disequilibrium [LD], which can be quantified at a population level). p-Value thresholding ($P_T$) was utilised for the gene test statistic calculation; four p-values were selected: all SNPs, $P_T < 0.5$, $P_T < 0.05$, and $P_T < 0.005$, meaning only SNPs below these thresholds were included in the gene-based model. We argue that distinct biological processes in individuals may only be captured when the optimal spectrum of polygenic variation is included in the model. A variety of $P_T$ could be utilised; for simplicity, we selected the four p-values thresholds described, as per our previous work (*Reay et al., 2020*). We mapped variants to 18297 autosomal genes in hg19 assembly defined by NCBI and obtained from the MAGMA website – genes within the MHC were removed due to the complexity of LD within this region. The 1000 genomes phase 3 European reference panel was utilised to define LD for input into MAGMA. Genic boundaries were extended to capture regulatory variation, with both conservative and liberal upstream and downstream boundary definition implemented. An extension of 5 kb upstream of the gene and 1.5 kb downstream was the conservative construct, whilst a larger 35 kb upstream and 10 kb downstream was the liberal construct. Boundaries were longer upstream of the gene in both instances to capture more promoter-related variation, as is usual practice (*Wray et al., 2018*; *Kunkle et al., 2019*; *Reay and Cairns, 2020*).

Genic p-values were transformed to $Z$-scores with the probit function for input into the gene-set association model. Competitive gene-set association was undertaken by a linear regression model whereby genic $Z$-scores are the outcome and confounders including gene size and genic minor allele count included as covariates. When these models are constructed at different $P_T$, this approach

constitutes testing whether the gene-set is more associated than the other genes, for which test statistics were calculated only including SNPs below the threshold. We selected pathways that survived multiple testing correction for an enrichment of lung function-associated variation relative to all other genes at that threshold by applying correction via the Benjamini–Hochberg (BH) method (FDR < 0.05) to all thresholds combined. These associations can be interpreted based on the p-value threshold for the model, for example, at gene-set which survives FDR correction that includes only variants which displayed a nominally significant univariable association with lung function (p<0.05) is indicative of a set of genes that are more associated with lung function than all other genes with at least one SNP that had p<0.05 in the GWAS. The BH approach was implemented rather than Bonferroni as several gene-sets will be tested multiple times at different p-value thresholds, and thus, the assumption of independence underlying Bonferroni correction likely means this would be overly conservative. In summary, gene-based test statistics were constructed at four different p-value thresholds, whereby only SNPs below the said threshold were included in the gene-based test statistic. Thereafter, competitive gene-set association is conducted for each druggable pathway at the different thresholds, with the null hypothesis being that the druggable pathway is no more associated with the trait (enriched with association) than all other genes for which gene-based p-values could be calculated by virtue of having an SNP below the threshold annotated to it. The concept underlying this is that distinct pathways may be enriched with common variants at differing levels of the polygenic signal, for example, a model including all SNPs will identify gene-set enriched with association relative to all other genes, whilst a less polygenic model, like a threshold of p<0.05, will capture gene-sets enriched with association relative to genes with at least one SNP mapped to it with a univariate association p<0.05. We defined gene-sets with known drug targets by sourcing hallmark and canonical (BioCarta, KEGG, PID, and Reactome) from the Molecular Signatures Database (MSigDB) (*Liberzon et al., 2015*) and retaining those with at least one gene with a high-confidence interaction with at least one approved pharmacological agent ($T_{Clin}$ genes), as annotated using the Target Central Resource Database (TCRD v6.1, $N_{Genes}$ = 613) (*Oprea et al., 2018*).

## PES candidate gene-set drug repurposing

We tested each candidate PES gene-set for overrepresentation of DrugBank compound targets using WebGestaltR v0.4.2 (*Liao et al., 2019*). Compounds were retained for each pathway if they survived FDR correction ($q < 0.05$) and were FDA approved. Single-drug gene matching was performed using the DGIdb v.3.02, with a minimum of three lines of supporting evidence the criterion for selection (*Cotto et al., 2018*).

The list of FDA-approved DrugBank compounds which were overrepresented targets in a PES candidate gene-set was reviewed by a pharmacist to prioritise potential useful compounds for lung function. A total of eight topical compounds were excluded. The remaining 55 oral and/or parenteral compounds were investigated for lung function-related adverse events (including all of dyspnoea, abnormal breath sounds, decreased respiratory rate, orthopnoea, shallow breathing, respiratory distress, respiratory depression, or any other related term), other alarming adverse events, important precautions, black-box warnings, or any contraindication that might prohibit the drug use in our study population. These data were reviewed for each compound using the following databases: drugs.com, Medscape, SIDER v4.1, and the summaries of each product's characteristics. We also searched for articles that discussed either an improvement or worsening in the lung functions for each compound along with the allowed paediatric age use.

The drugs were then categorised into one of five categories (*Figure 3—figure supplement 1*, *Supplementary file 2d*). Level 1 was assigned for an oral or parenteral formulation, with no documented respiratory side effects and with positive evidence of prior use for lung function in the literature. Level 2 was assigned for an oral or parenteral formulation, with no documented or rare (<1%) respiratory side effects and with/without positive evidence but no negative evidence of prior use for lung function in the literature. Level 3 was assigned to an oral or parenteral formulation, with common (1–15%) respiratory side effects and with/without positive evidence but no negative evidence of prior use for lung function in the literature. Level 4 compounds were those oral or parenteral formulations with very common (16–50%) respiratory side effects or other alarming adverse effects unrelated to respiratory function, without positive evidence but with/without negative evidence of prior use for lung function in the literature. Finally, level 5 was assigned when the drug was associated with a serious adverse event (including a black-box warning or an absolute contraindication).

## The PES model for individuals

We defined the model to calculate PES profiles for individuals as follows (Equation 1). Consider $j$ SNPs for $i$ individuals, wherein the SNPs are those physically mapped to genes which are members of a candidate PES gene-set ($m$). Let $\beta_j$ denote the statistical effect size for each variant from the GWAS, multiplied by its dosage $G_{ij}$. The SNPs included were those below the p-value threshold utilised to discover the gene-set.

$$PES_i = \sum_{i=1}^{m} \hat{\beta}_j G_{ij} \tag{1}$$

We averaged these scores by the number of SNPs carried by each individual and scaled them using the *scale*() function in R. PES profiles were generated in all instances by first filtering the GWAS summary statistics for common variants (MAF >0.01) within the genic boundaries of variants which comprise the PES gene-set. The genic boundaries were extended using the liberal or conservative configuration, dependent on which boundary definition was utilised in the gene-set association for that pathway. PRSice v2.2.12 calculated the respective PES, along with genome-wide PGS (using the same additive model but genome wide) for FEV₁ and FVC (*Choi and O'Reilly, 2019*).

## Lung function PES in the HCS cohort

We utilised an independent, genotyped cohort for which spirometry measures were recorded to investigate the phenotype relevance of PES profiles for lung function. Participants were drawn from the HCS, a population-based cohort of individuals aged between 55 and 85 years, predominantly of European ancestry and residing in Newcastle, New South Wales, Australia. All work was conducted in accordance with ethics committee approvals. Consenting participants completed a series of questionnaires, attended a clinic visit, and provided blood samples. Individuals were recruited by random selection from the New South Wales State electoral roll with detailed recruitment and data collection methods for the HCS described elsewhere (*McEvoy et al., 2010*). Participants provided blood samples from which DNA was extracted and genotyped using the Affymetrix Axiom Kaiser array. Quality control excluded SNPs with genotype call rate of <0.95, deviation from Hardy–Weinberg equilibrium ($p<1\times10^{-6}$) or MAF of <0.01. The input for relatedness testing and removal of population outliers were autosomal, common (MAF >0.05), physically genotyped SNPs in relative linkage equilibrium ($r^2$ <0.02), with regions of long-range LD removed, as is usual practice (*Price et al., 2008*). We used PLINK 1.9 to retain only unrelated individuals (pi_hat >0.185), with one participant from each related pair blinded to phenotype information. Population outliers were determined by performing principal component analysis (PCA) using PLINK 1.9. We clustered individuals in the HCS with the first two principal components from each 1000 genomes phase 3 superpopulation using *k*-means clustering. Thereafter, we conservatively excluded any HCS individual with a first or second principal component above or below the maximum or minimum 1000 genomes European values for these eigenvectors. PCA was repeated in the filtered European ancestry HCS subset such that eigenvectors could be used as downstream covariates. Imputation to the Haplotype Reference Consortium panel involved a series of steps and additional data clean up, reference lift over to the hg19/GRCh37, and data submission to the Michigan imputation server, as specified in the submission guidelines (*Loh et al., 2016*; *Das et al., 2016*). Post-imputation quality control was as follows: imputation $R^2$ >0.8, MAF >0.01, and missingness <0.02. We retained common variants (MAF >0.01) with high imputation quality ($R^2$ >0.8).

Spirometry data from the HCS was then processed by selecting individuals with non-missing FEV₁ and FVC. We utilised the maximum FEV₁ and FVC from four attempts and fitted a linear model which covaried for sex, age, age², height, height², smoking status, self-reported asthma status, and self-reported bronchitis/emphysema status. The phenotype for association testing was residuals from these models transformed via inverse-rank normalisation (Blom transformation) using the RNOmni package. We tested the association between a genome-wide PGS for FEV₁ and FVC ($P_T$ <1, 0.5, 0.05, 0.005, $5 \times 10^{-5}$, $5 \times 10^{-8}$) with their respective transformed spirometry indices adjusted for the first five SNP-derived principal components using PRSice v2.2.12. Similarly, the association between each of the PES profiles with an overrepresentation of FDA-approved drug targets and FEV₁ and/or FVC was investigated using the same approach; however, we only constructed the

PES at the p-value for which it demonstrated the strongest association signal after multiple testing correction in the GWAS. We further adjusted each of these models for genome-wide PGS at the same $P_T$ for which the PES was calculated.

## The relationship between PES and mRNA expression

We obtained RNAseq normalised read counts (PEER normalised RPKM) for 23723 genes which survived QC in the Geuvadis dataset (https://www.ebi.ac.uk/arrayexpress/experiments/E-GEUV-1/files/analysis_results/?ref=E-GEUV-1). The Geuvadis project performed RNAseq on transformed LCL for participants in the 1000 genomes project (*Lappalainen et al., 2013*). We retained 357 European individuals in this dataset for which phase 3 sequencing data was available from the 1000 genomes. The association between normalised mRNA expression for genes part of the candidate gene-set and each PES was tested using a linear model, adjusted for sex, the first three SNP-derived principal components, and genome-wide PGS at the same $P_T$ utilised to calculate the PES. Multiple testing correction was applied for the number of genes in each set via the BH method using the *p.adjust()* function.

## Transcriptome-wide association studies

A TWAS of each lung function measure was performed using the FUSION software (*Gusev et al., 2016*). SNP weights were derived for genes with a significant contribution of *cis* acting SNPs to expression variability (*cis-h²p*<0.01) using lung and whole blood RNAseq GTEx v7 data. A transcriptome-wide significant gene was defined by accounting for the number of genes with models of genetically regulated expression in lung and whole blood, respectively – lung: p<6.43×10⁻⁶(α = 0.05/7776); whole blood: p<8.32×10⁻⁶(α = 0.05/6007). A more conservative threshold could be applied which corrects for all models in both tissues (p<3.6×10⁻⁶); however, given the correlation between models and the discovery nature of this study, we chose the more liberal correction threshold. We excluded genes within the MHC region due to its LD complexity. Furthermore, we subjected two smoking behaviour phenotypes to TWAS to uncover associations which could be driven by residual effects of smoking. This is inherently conservative as it is possible that genes associated with both lung function and smoking behaviours could exhibit pleiotropic effects; however, as we wish to define drug targets relevant to lung function, the exclusion of these shared genes is warranted. The smoking phenotypes were 'ever vs. never smoked' and 'cigarettes smoked per day', and TWAS was performed using lung and blood for consistency, along with SNP weights from the dorsolateral prefrontal cortex and nucleus accumbens, as these brain regions have been implicated in nicotine addiction. Genes which survived the above were searched using DGIdb, with the following criteria utilised to define gene-target pairs, where the drug mode of action matched the sign of the TWAS *Z* value: (i) tier 1 – FDA-approved compound with at least two lines of evidence for interacting with the target gene; (ii) tier 2 – investigational compound (not FDA approved) with at least two lines of evidence for interacting with the target gene.

The TWAS Miami plots were generated using an edited version of the TWAS-plotter.V1.0.R script (https://github.com/opain/TWAS-plotter). A Bayesian method FOCUS was utilised to finemap the TWAS associations which could be therapeutically useful (tier 1 or 2) (*Mancuso et al., 2019*). Given observed TWAS statistics, the marginal posterior inclusion probability (PIP) was calculated and subsequently used to compute a credible set with 90% probability ($\rho$) of containing the causal gene ($c_i = 1$). As FOCUS allows the null model to be predicted as a possible member of the credible set, we excluded any genes for which that occurred. The credible set ($S$) was defined by summing normalised PIP such that $\rho$ was exceeded, sorting the genes and then including those genes until at least $\rho$ of the normalised-posterior mass is explained (Equation 2).

$$S\{Gene_1, \ldots, Gene_k\} = \sum_{i=1}^{k} PIP\,(c_i = 1|Z_{TWAS}) \geq \rho \qquad (2)$$

The Bernoulli prior for each causal indicator was set as the default $p = 1\times10^{-3}$, with a default prior variance for effects at causal genes set as 40 ($n\sigma_c^2=40$). Previous work has demonstrated that FOCUS-computed PIPs were robust to different specified prior variances (*Mancuso et al., 2019*); however, we further utilised a more conservative prior of $p = 1\times10^{-5}$ to assess the effect on the *PIP* calculated for candidate druggable genes. In all instances, we utilised a multi-tissue panel obtained

from FOCUS GitHub repository which combines GTEx v7 SNP-weights with other FUSION TWAS weights (https://github.com/bogdanlab/focus/wiki, GTEx v7 with METSIM, CMC, YFS, and NTR). The marginal TWAS $Z$ to use for finemapping for each locus was selected in the tissue for which the gene was found to be associated via the FUSION TWAS methodology (lung or blood), if available, otherwise by predictive accuracy (cross-validated $R^2$).

### Host–viral interactome data

We selected three respiratory viruses for which host–viral protein interaction data was previously published: SARS-CoV2, H1N1, and the HAdV family. The host–SARS-CoV2 interactome was defined using affinity-purification mass spectrometry ($N_{Genes}$ = 332, MiST score $\geq$0.7, a SAINTexpress BFDR $\leq$0.05) (*Gordon et al., 2020*). We selected 91 proteins which both interact with viral proteins expressed by influenza (mass spectrometry) and siRNA-mediated downregulation-reduced viral replication in cultured cells by at least three $\log_{10}$ units while retaining >80% cell viability (*Watanabe et al., 2014*). Finally, the HAdV–host interactome was defined using a protein microarray platform ($N_{Genes}$ = 24), which encompasses 20 viral proteins encoded by five HAdV species (*Martinez-Martin et al., 2016*). We investigated approved inhibitors or antagonists of these genes using DGIdb as described above in the PES candidate gene-set drug repurposing section. The sets of genes which interact with viral proteins for each virus ('viral prey proteins') were subjected to over-representation analysis using the GENE2FUNC function of FUMA (*Watanabe et al., 2017*). We selected gene-sets which survived multiple testing correction ($q < 0.05$), which contained at least one of the following key terms related to glycaemic biology: glucose, insulin, diabetes, or glucagon. Further, we investigated whether there was a significant overrepresentation of interactions amongst these viral prey proteins overlapping a glycaemic pathway using STRING v11.0 (*Szklarczyk et al., 2019*). We assembled a list of antidiabetic drug targets by searching compounds annotated with the level 2 ATC code A10 (drugs used in diabetes) in DGIdb, retaining drug–gene interactions with two or more lines of evidence. The interactions between these drug target proteins and the glycaemic SARS-CoV2 prey proteins were investigated once more using STRING, with only interactions scoring >0.75 considered.

## Acknowledgements

This study was supported by an NHMRC project grant (1147644). MJC is supported by an NHMRC Senior Research Fellowship (1121474).

Hunter Cohort Study: The research on which this paper is based was conducted as part of the Hunter Community Study, The University of Newcastle. We are grateful to the University of Newcastle for funding and to the men and women of the Hunter region who provided the information recorded.

## Additional information

### Competing interests

William R Reay, Murray J Cairns: has filed a patent related to the use of the pharmagenic enrichment score methodology in complex disorders. This competing interest only applies to that section of the manuscript. WIPO Patent Application WO/2020/237314. The other authors declare that no competing interests exist.

### Funding

| Funder | Grant reference number | Author |
| --- | --- | --- |
| National Health and Medical Research Council | 1147644 | Murray J Cairns |
| National Health and Medical Research Council | 1121474 | Murray J Cairns |

The funders had no role in study design, data collection and interpretation, or the decision to submit the work for publication.

## Author contributions
William R Reay, Conceptualization, Data curation, Software, Formal analysis, Investigation, Visualization, Methodology, Writing - original draft, Project administration, Writing - review and editing; Sahar I El Shair, Formal analysis, Writing - review and editing; Michael P Geaghan, Software, Formal analysis, Writing - review and editing; Carlos Riveros, Data curation, Formal analysis, Writing - review and editing; Elizabeth G Holliday, Resources, Data curation, Formal analysis, Writing - review and editing; Mark A McEvoy, Stephen Hancock, Roseanne Peel, Rodney J Scott, John R Attia, Resources, Data curation, Writing - review and editing; Murray J Cairns, Conceptualization, Resources, Data curation, Supervision, Funding acquisition, Visualization, Writing - original draft, Project administration, Writing - review and editing

## Author ORCIDs
William R Reay https://orcid.org/0000-0001-7689-2453
Murray J Cairns https://orcid.org/0000-0003-2490-2538

## Ethics
Human subjects: The use of the Hunter Community Cohort data was approved by the University of Newcastle Human Ethics Research Committee (HREC, reference: H-820-0504a). All other information related to ethical approval for the individual GWAS studies we utilised in this study are detailed in their respective publications as referenced throughout the text.

## Decision letter and Author response
Decision letter https://doi.org/10.7554/eLife.63115.sa1
Author response https://doi.org/10.7554/eLife.63115.sa2

# Additional files

## Supplementary files
• Supplementary file 1. Genetic correlation and causal inference results. (a) Genetic correlations between $FEV_1$ and 172 GWAS from the LDhub library. (b) Genetic correlations between FVC and 172 GWAS from the LDhub library. (c) Genetic correlations between $FEV_1$/FVC and 172 GWAS from the LDhub library. (d) Latent casual variable models constructed between metabolic/hormonal traits and measures of lung function. (e) Two-sample Mendelian randomisation models testing the effect of fasting glucose on lung function. (f) Two-sample Mendelian randomisation models testing the effect of fasting glucose on lung function: sensitivity and pleiotropy analyses. (g) Two-sample Mendelian randomisation models testing the effect of fasting glucose on lung function: leave-one-out analyses. (h) Phenome-wide association study of rs17168486 using GWAS atlas summary statistics. (i) Phenome-wide association study of rs7903146 using GWAS atlas summary statistics. (j) Phenome-wide association study of rs6975024 using GWAS atlas summary statistics. (k) Phenome-wide association study of rs1260326 using GWAS atlas summary statistics. (l) Phenome-wide association study of rs560887 using GWAS atlas summary statistics. (m) Association of fasting glucose IV SNPs with smoking behaviour phenotypes – ever vs. never smoked. (n) Association of fasting glucose IV SNPs with smoking behaviour phenotypes – cigarettes smoked per day.

• Supplementary file 2. Pharmagenic enrichment score gene-sets. (a) Candidate PES gene-set identification for $FEV_1$ and FVC – FDR < 0.1 (conservative genic boundaries). (b) Candidate PES gene-set identification for $FEV_1$ and FVC – FDR < 0.1 (liberal genic boundaries). (c) FDA-approved DrugBank compounds with overrepresented targets in a PES candidate gene-set. (d) Evaluation of FDA-approved DrugBank compounds with overrepresented targets in a PES candidate gene-set. (e) Single drug–gene interactions in the ALK pathway gene-set. (f) Single drug–gene interactions in the basal cell carcinoma gene-set. (g) Single drug–gene interactions in the dilated cardiomyopathy

gene-set. (h) Single drug–gene interactions in the Hedgehog signalling pathway gene-set (KEGG). (i) Single drug–gene interactions in the pathways in cancer gene-set (j) Single drug–gene interactions in the TGF-beta signalling pathway gene-set. (k) Single drug–gene interactions in the ECM (NABA matrisome) gene-set. (l) Single drug–gene interactions in the BMP pathway gene-set. (m) Single drug–gene interactions in the Hedgehog signalling pathway (PID) gene-set. (n) Single drug–gene interactions in the circadian clock gene-set. (o) Single drug–gene interactions in the class b2 secretin family receptors gene-set. (p) Single drug–gene interactions in the extension of telomeres gene-set.

• Supplementary file 3. Pharmagenic enrichment score profiles and transcriptome-wide association study results. (a) Genome-wide polygenic score (PGS) of lung function in the Hunter Community Study cohort. (b) Pharmagenic enrichment scores tested in the Hunter Community Study cohort (unadjusted for genome-wide PGS). (c) Association of pharmagenic enrichment scores (PES) with lung function in the Hunter Community Study cohort – covariation for genome-wide PGS. (d) Correlation between the $FEV_1$ pathways in cancer PES and gene expression in the Geuvadis cohort. (e) Correlation between the FVC pathways in cancer PES and gene expression in the Geuvadis cohort. (f) Correlation between the FVC ECM PES and gene expression in the Geuvadis cohort. (g) Correlation between the $FEV_1$ extension of telomeres PES and gene expression in the Geuvadis cohort. (h) Correlation between the $FEV_1$ dilated cardiomyopathy PES and gene expression in the Geuvadis cohort. (i) Correlation between the FVC class b2 secretin PES and gene expression in the Geuvadis cohort. (j) Correlation between the FVC circadian clock PES and gene expression in the Geuvadis cohort. (k) Transcriptome-wide association study (TWAS) of $FEV_1$ using lung tissue-derived SNP weights. (l) Transcriptome-wide association study (TWAS) of FVC using lung tissue-derived SNP weights. (m) Transcriptome-wide association study (TWAS) of $FEV_1$ using whole blood-derived SNP weights. (n) Transcriptome-wide association study (TWAS) of FVC using whole blood-derived SNP weights. (o) Transcriptome-wide association study (TWAS) of cigarettes per day using whole blood-derived SNP weights. (p) Transcriptome-wide association study (TWAS) of ever vs. never smoked using whole blood-derived SNP weights. (q) Transcriptome-wide association study (TWAS) of cigarettes per day using brain (DLPFC)-derived SNP weights. (r) Transcriptome-wide association study (TWAS) of ever vs. never smoked using brain (DLPFC)-derived SNP weights. (s) Transcriptome-wide association study (TWAS) of cigarettes per day using lung tissue-derived SNP weights. (t) Transcriptome-wide association study (TWAS) of ever vs. never smoked using lung tissue-derived SNP weights. (u) Transcriptome-wide association study (TWAS) of cigarettes per day using brain (Nucleus accumbens)-derived SNP weights. (v)Transcriptome-wide association study (TWAS) of ever vs. never smoked using brain (Nucleus accumbens)-derived SNP weights. (w) Candidate drug–gene interactions with lung function increasing potential based on the sign of the TWAS Z value.

• Supplementary file 4. Host-viral interactome results. (a) SARS-CoV2 host–viral interactome – human genes which putatively interact with a viral SARS-CoV2 protein (Gordon et al.). (b) Influenza host–viral interactome – human genes which putatively interact with a viral influenza protein (Watanabe et al, top 91 interacting proteins). (c) Human adenovirus host–viral interactome – human genes which putatively interact with a viral adenovirus protein (Martinez-Martin et al). (d) SARS-CoV2 viral prey proteins which overlap a glycaemic gene ontology pathway. (e) SARS-CoV2 viral prey proteins overlapping glycaemic pathways that demonstrate a high-confidence interaction with an antidiabetic drug target. (f) List of putative antidiabetic compound target genes (DGIdb).

• Transparent reporting form

### Data availability

All data are publicly available from the references described in the manuscript. Code related to this study can be found at the following link: https://github.com/Williamreay/Lung_function_drug_repurposing_manuscript copy archived at https://archive.softwareheritage.org/swh:1:rev:01aef11a0cc0c7f897c9497126d2ce454108eff1/.

The following previously published datasets were used:

| Author(s) | Year | Dataset title | Dataset URL | Database and Identifier |
|---|---|---|---|---|
| Shrine N, Guyatt | 2019 | GWAS summary statistics - lung | ftp://ftp.ebi.ac.uk/pub/ | GWAS Catalog, |

| AL, Erzurumluoglu AM, Jackson VE, Hobbs BD, Melbourne CA | | function | databases/gwas/summary_statistics/ShrineN_30804560_GCST007432 | GCST007432 |
| --- | --- | --- | --- | --- |
| Lappalainen T, Sammeth M, Friedländer MR | 2019 | GEUVADIS | https://www.ebi.ac.uk/arrayexpress/experiments/E-GEUV-1/files/analysis_results/?ref=E-GEUV-1 | ArrayExpress, E-GEUV-1 |

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
