## [Decision Letter]

**Acceptance summary:**

This paper is of interest to researchers seeking to use genetics in order to reposition drugs that improve lung function. The work highlights biochemical traits that could be targeted to modulate lung function. The analyses have been performed to a high level, with some of the most interesting and novel results being of modest statistical significance.

**Decision letter after peer review:**

[Editors’ note: the authors submitted for reconsideration following the decision after peer review. What follows is the decision letter after the first round of review.]

Thank you for submitting your work entitled "Genetic association and causal inference converge on hyperglycaemia as a modifiable risk factor for respiratory disease" for consideration by *eLife*. Your article has been reviewed by three peer reviewers, and the evaluation has been overseen by a Reviewing Editor and a Senior Editor. The following individual involved in review of your submission has agreed to reveal their identity: Bogdan Pasanuic (Reviewer #3).

Our decision has been reached after consultation among all the reviewers. Based on these discussions and the individual reviews below, we regret to inform you that your work will not be considered further for publication in *eLife*.

Reviewers considered that the manuscript's focus on rapidly repositioning drugs to improve lung function is timely. They, however, remained unconvinced by some of the main claims and their novelty, and expressed concerns regarding the robustness of results that are based on border-line statistics. Replication and/or validation of the main results might have helped to convince the reviewers. Further consideration of other explanatory variables would also have provided greater robustness to the claims of causality.

Reviewer #1:

This paper seeks to use genetics to reposition drugs to improve lung function. A new pipeline is used to connect together several recent genetics methods: filtering traits on genetic correlations; refining with causality tests (LCV+MR); and then testing lung function and gene expression with PES and TWAS.

Overall, I am quite positive on the goals and core ideas in this paper. The focus on quickly repositioning drug to improve lung function is timely. However, the most interesting and novel results are statistically borderline. Methodologically, I do not see anything new in the paper.

1) There is not much evidence that the PES have enriched signal for lung function (Table 3). The primary PES analyses do not adjust for the overall PRS, hence do not establish Pharmagenic Enrichment, a PRS built on a random subset of SNPs would be expected to have a nonzero effect. The authors recognize this and perform secondary analyses conditional on the PRS to test for enrichment; however, this analysis has null results (Bonferroni-adjusted across 8 tests gives minimum p>.2).

(1a) I don't see any explanation of the permutation test used here, but the details wrt multiple PES thresholds and whether covariates and PRS are permuted as well as the PES are essential and can significantly change results.

2) The PYGB finding is very nice, but was previously found in a similar analysis (in the paper producing the summary stats used here, see Table 1 in Shrine et al., 2019, PMC6397078). This paper also identified the TGF-β superfamily signalling pathway. The observation that this gene can be putatively targeted by Sivelestat is novel (as far as I know) and potentially very exciting, however, this is not discussed much, and no validation is given for the gene-drug interaction, and no explanation is given to relate neutrophil elastase to glucose.

3) I believe the covid analysis assesses only glycemic pathways (Table 4), hence it is hard to evaluate whether the “prey proteins” are more enriched in glycemic pathways than in any other biologically meaningful pathways (further, in the Discussion, it is said that these genes are very pleiotropic). In the future, I think this analysis could be strengthened by testing the PES (or ordinary PRS) against measurements of these proteins in healthy samples, which would demonstrate the link from druggable (or general) glucose biology to the covid-relevant proteins. However, nontrivial effort would be required to integrate such pQTL summary statistics, though I believe such datasets are freely available..

(4a) The LCV paper recommends considering only tests with |GCP|>.6, this rules out the LCV test for FEV1-glucose, FEV1/FVC-HDL, or FVC-leptin. If there is a reason to deviate from the recommended practice, it should be explained.

(4b) Likewise, the MR analyses have only a very weak statistical signal (p=.02,.03): 1 this doesn't survive correction for testing two phenotypes (not to mention the implicit tests prioritized by rhog that were discarded based on LCV); 2 the LCV paper proves these tests are susceptible to inflation by genetic correlation; 3 I do not agree that horizontal pleiotropy has been ruled out, a priori it seems almost certain that many heritable traits (BMI, smoking, diet, exercise,.…) will causally effect both glucose and lung function, to some extent, and moreover you do show that AMT has near-significant TWAS effects on both smoking and glucose.

Reviewer #2:

In this study, Reay et al. used publicly available GWAS data with regards to lung function and biochemical traits to identify molecular mechanisms capable of improving lung function. There are several comments and questions:

1) Reay et al. identified multiple biochemical traits that could be targeted in order to modulate lung function including fasting glucose and fasting insulin levels as well as other glycaemic related pathways and traits. Of these traits they also identify four gene-sets overrepresented with proteins that interact with viral SARS-CoV2 proteins. However as mentioned by Reay et al., previous studies have already found glycaemic control in the form of diabetes to have an effect on both lung function [Klein et al. Diabet Med. 2010] as well as Covid-19 risk and severity [Yang et al., Int J Infect Dis. 2020 94:91-95.]. The results presented here are not groundbreaking on their own.

2) In addition, the authors have employed several statistical methods using existing datasets and performed a comprehensive analysis. However, all of the approaches are from literature, which limits the novelty of this study.

3) The benefit of using the framework proposed by Reay et al. is that it identifies potential new uses for existing drugs through the biochemical traits they modulate. This however means that the potential discoveries regarding drug repurposing are limited to only those compounds with known biochemical effects. Another limitation is the use of genetics data exclusively and not integrating more layers of information that might identify causal traits for any given disease. Various other approaches have extensively been reported before (e.g. Pushpakom et al. Nat Rev Drug Discov. 2019) which creates the question as to how this methodology can be edited in order to maximize the possible findings.

4) The authors claimed that "The correlation between the expression of genes within each pathway encompassed by the PES and the PES profiles themselves could provide further support for their biological impact". This is true when the expression data of the genes come from the relevant tissues. Here, the authors focus on lung function but performed "association between lung function PES and gene expression using RNA sequencing (RNAseq) on transformed lymphoblastoid cell lines (LCL)". My question is how is LCL relevant for lung function?

5) Another question is as to how these findings can be validated either in vivo or in vitro. Figure 5 shows a schematic representation on how treatment could be implemented but it is unclear if any validation experiments have been performed.

6) Throughout this study, the authors used three measurements of spirometry phenotypes for lung function. Then, the results and interpretation in this study should be limited to "lung function". However, the authors generalized their observations from "lung function" to "respiratory disease and respiratory infection". This can be misleading (too far-reaching). For example, lung function is often measured by dynamic spirometry which mostly reflects large airway function. However, respiratory disease like COPD is an inflammatory airway disease which affects the small airways in particular (DS Postma, NJEM, ‎2015). Furthermore, "respiratory infection" by bacterial and viral infection such as tuberculosis, influenza and coronaviruses may lead to completely different pathogenesis. It is hard to believe that hyperglycaemia will have causal effect on these respiratory diareses (infections).

7) For all candidate genes identified by TWAS analysis, they could be further prioritized by checking if they are differentially expressed in the lung between samples with and without impaired lung functions.

8) The authors stated "Probabilistic finemapping of these transcriptome-wide significant regions using a multi-tissue reference panel was then performed to prioritize whether these genes are likely causal at that locus". What is the reasoning behind the prioritization using “multi-tissue reference”? It is known that the majority of cis-eQTLs are shared across tissues, but how this could help to prioritize relevant genes?

Reviewer #3:

The manuscript by Reay et al. presents a set of comprehensive analyses of GWAS data to postulate the causal role of hyperglycemia in lung function. The authors perform a series of causal inference analyses on the GWAS data of several blood traits to identify genetically correlated traits that can be explained by a causal role; the authors then seek to identify drug repurposing targets through two complementary analyses, a polygenic risk score restricted to regions within druggable targets and a transcriptome-wide scan linking genetically predicted expression in blood and lung tissue to lung function. Overall, the manuscript leverages recently introduced sophisticate statistical methods and does a thorough job in stress testing the findings. The putative causal role of fasting glucose joint with putative target genes is an important addition to the field. My main comments relate to the robustness of the causal claims.

1) The MR analyses assume the blood traits (i.e. fasting glucose) are mediating lung function. Whereas several biological plausible avenues are given in the discussion for this assumption, it can certainly be the case that lung function is mediating fasting glucose (e.g., lung function causing overall body impairment which in turn causes changes in blood measurements). I strongly encourage the authors to perform analyses under this reverse causality assumption. In particular, the bivariate MR method of Pickrell NG 2016 would be relevant here.

2) As the authors describe in the Discussion section, wrong assumptions in the MR framework can invalidate the findings. The authors do a great job in assaying the impact of pleiotropy on the MR estimates using recently developed methods (LCV, MR-PRESSO etc); however the causal role of smoking is left ambiguous in the causal inference. Clearly smoking has a causal role on lung function, and GWAS of smoking reveals genetic correlates of smoking status (amount). Is there any impact of smoking on blood traits? Is smoking a collider in the causal diagram genetics -> fasting glucose -> lung function? The authors have access to GWAS of smoking and could leverage the MR toolkit to investigate causal effects of smoking on glucose.

3) The identification of drug repurposing tools using the PES score is inconclusive without some replication/validation. The PES is explaining a small proportion of variation in the trait making the interpretation of PES correlations subtle at best; e.g., it is hard to find a biological role for some of the gene-sets that show significance in Table 2. More importantly, it is unclear what is the null expectation of the PES-gene expression correlation analysis; that is, if PES is computed using random pathways (i.e. not specific to druggable pathways) and re-runs the analyses, what are the results? Or, reversely, if the authors perform the same analyses for a randomly chosen complex trait (e.g., height/bmi), what pathways show up in Tables 2/3?

[Editors’ note: further revisions were suggested prior to acceptance, as described below.]

Thank you for submitting your article "Genetic association and causal inference converge on hyperglycaemia as a modifiable risk factor for respiratory disease" for consideration by *eLife*. Two of the three reviewers provided further comments on your manuscript and there followed extensive discussion among them and *eLife* Editors. We note that the third reviewer previously shared concerns of robustness of results based on border-line statistics and requested that the causality analysis needed to be improved. The evaluation was overseen by a Reviewing Editor and David James as the Senior Editor. The Reviewing Editor has drafted this decision to help you prepare a revised submission.

We would like to draw your attention to changes in our revision policy that we have made in response to COVID-19 (https://elifesciences.org/articles/57162). Specifically, we are asking editors to accept without delay manuscripts, like yours, that they judge can stand as *eLife* papers without additional data, even if they feel that they would make the manuscript stronger. Thus the revisions requested below only address clarity and presentation. Nevertheless, the reviewers and editors emphasise that as currently written the work is not yet ready for publication.

Summary:

The authors used publicly available GWAS data with regards to lung function and biochemical traits seeking to use genetics to reposition drugs to improve lung function. The authors sought to identify drug repurposing targets through two complementary analyses: a polygenic risk score restricted to regions within druggable targets and a transcriptome-wide scan linking genetically predicted expression in blood and lung tissue to lung function. Reviewers valued the sophisticated approaches applied in this analysis although questioned the authors' statistical interpretations noting that some results were borderline significant.

Revisions:

The joint view of the reviewers and *eLife* Editors is that several of the reviewers' comments have been addressed adequately. Nevertheless, other reviewers' comments were not resolved by your revision, most importantly Points 1 and 4 of Reviewer 1. As a group we would be more supportive of the revision if the causality claims were to be toned down with appropriate caveats included throughout, and if the statistical results were to be more appropriately presented. The current version, without such changes, is not judged to be ready for publication.

We are content that you now acknowledge the LCV publication's recommendation that only tests with |GCP| > 0.6 are considered. However, your revision does not follow this recommendation unswervingly: e.g. "We acknowledge that the posterior mean GCP estimate for the FEV1 does not quite the threshold of > 0.6, and thus, the causal relationship was more rigorous with FVC". Moreover, your revision unnecessarily clouds this important issue: "|GCP| > 0.6 previously postulated to be evidence of a rigorous relationship". We do not support uneven application of an established threshold.

It is now acknowledged that some tests were not significant after correcting for multiple tests. Nevertheless, an unwarranted emphasis was sometimes placed on non-significant results, e.g. "However, this still suggested that there was a relationship between the Class B/2 secretin family receptor FVC PES and FVC beyond what is attributable to a genome-wide PGS" and "several gene-sets trended towards surviving correction". Similar problems identified among the MR tests and the adaptive choices for PRS p-value thresholds still need to be addressed. The MR and PES results have relatively weak statistical support and yet this is not reflected by an emphasis placed on them in the Abstract. We are of the view that marginally significant (or null) results can still provide a significant contribution to the field as long as their statistical support is reported appropriately. The manuscript will require changes to the application and interpretation of statistical tests throughout.

---

## [Author Response]

[Editors’ note: The authors appealed the original decision. What follows is the authors’ response to the first round of review.]

Reviewers considered that the manuscript's focus on rapidly repositioning drugs to improve lung function is timely. They, however, remained unconvinced by some of the main claims and their novelty, and expressed concerns regarding the robustness of results that are based on border-line statistics. Replication and/or validation of the main results might have helped to convince the reviewers. Further consideration of other explanatory variables would also have provided greater robustness to the claims of causality.Reviewer #1:This paper seeks to use genetics to reposition drugs to improve lung function. A new pipeline is used to connect together several recent genetics methods: filtering traits on genetic correlations; refining with causality tests (LCV+MR); and then testing lung function and gene expression with PES and TWAS.Overall, I am quite positive on the goals and core ideas in this paper. The focus on quickly repositioning drug to improve lung function is timely. However, the most interesting and novel results are statistically borderline. Methodologically, I do not see anything new in the paper.1) There is not much evidence that the PES have enriched signal for lung function (Table 3). The primary PES analyses do not adjust for the overall PRS, hence do not establish Pharmagenic Enrichment, a PRS built on a random subset of SNPs would be expected to have a nonzero effect. The authors recognize this and perform secondary analyses conditional on the PRS to test for enrichment; however, this analysis has null results (Bonferroni-adjusted across 8 tests gives minimum p>.2).

The reviewer is correct to point out that secondary analyses conditional on the overall genome-wide score for models testing PES are an important component. However, we would posit that this does sacrifice some power, particularly as some of the PES display a significant correlation with genome-wide lung function PGS. We agree that the association observed between the *Class b/2 secretin* PES and FVC was only nominal given the number of tests performed and state as such in the text. In accordance with the reviewer’s, comments we have edited that section of the manuscript to more clearly state the multiple testing burden associated with the results, such that it is more interpretable to the reader.

(1a) I don't see any explanation of the permutation test used here, but the details wrt multiple PES thresholds and whether covariates and PRS are permuted as well as the PES are essential and can significantly change results.

We agree with the reviewer that our description of the permutation terminology was somewhat lacking. In addition, upon reviewing the manuscript we decided that it would enhance the interpretability of our results to provide raw *P* values and correct for the number of tests using the Bonferroni method. The manuscript has now been adjusted accordingly. Importantly, this did not materially change which PES were significantly associated.

2) The PYGB finding is very nice, but was previously found in a similar analysis (in the paper producing the summary stats used here, see Table 1 in Shrine et al., 2019, PMC6397078). This paper also identified the TGF-β superfamily signalling pathway. The observation that this gene can be putatively targeted by Sivelestat is novel (as far as I know) and potentially very exciting, however, this is not discussed much, and no validation is given for the gene-drug interaction, and no explanation is given to relate neutrophil elastase to glucose.

We agree with the reviewer that *PYGB* has previously been linked to lung function via the GWAS performed in the Shrine *et al.* manuscript. However, we would argue that the finding in this study is novel given we are able to integrate gene expression data via TWAS and subsequent Bayesian finemapping to assign a direction of effect to this association, that is, downregulation was associated with increased FEV_1_. The finemapping procedure is particularly valuable in this situation given that we can be more confident in the robustness of the involvement of this gene in pulmonary biology. The putative interaction between *PYGB* and Sivelestat was derived using public databases as curated by DGIdb v.3.0.2, and we agree with the reviewer that the potential uncertainty of this interaction needs to be stated to the reader. We have edited the TWAS section of the manuscript as such to address this. Moreover, we have added that *PYGB* and the gene that encodes neutrophil elastase have a high confidence interaction as per the STRING database.

Furthermore, we acknowledge that the role of TGF-β signalling has been previously characterised in the literature in terms of its involvement in lung function. We would argue that using this pathway to construct a polygenic score that could inform drug repurposing via the *pharmagenic enrichment score* approach is the novel component of our study, as opposed to discovering genetic support for its role in the lung.

3) I believe the covid analysis assesses only glycemic pathways (Table 4), hence it is hard to evaluate whether the “prey proteins” are more enriched in glycemic pathways than in any other biologically meaningful pathways (further, in the Discussion, it is said that these genes are very pleiotropic). In the future, I think this analysis could be strengthened by testing the PES (or ordinary PRS) against measurements of these proteins in healthy samples, which would demonstrate the link from druggable (or general) glucose biology to the covid-relevant proteins. However, nontrivial effort would be required to integrate such pQTL summary statistics, though I believe such datasets are freely available..

The reviewer makes an important comment that the enrichment of “prey proteins” in the glycaemic pathways needs appropriate contextualisation. It should be noted that the glycaemic gene-sets survived multiple testing correction (*q* < 0.05) after testing for an overrepresentation of these genes in every gene-set featured by the GENE2FUNC tool in FUMA (thousands of pathways), and thus, whilst these pathways may not necessarily be the most meaningful gene-sets that display an overrepresentation of SARS-CoV2 “prey proteins”, their overrepresentation is statistically robust and relevant given our overarching results related to the importance of glycaemic biology to the lung. It is an excellent suggestion by the reviewer to test the association between PGS and/or PES with protein expression of SARS-CoV2 “prey proteins”. We believe that this could be an important future direction arising from this study and have included this in the discussion.

(4a) The LCV paper recommends considering only tests with |GCP|>.6, this rules out the LCV test for FEV1-glucose, FEV1/FVC-HDL, or FVC-leptin. If there is a reason to deviate from the recommended practice, it should be explained.

We agree with the reviewer, that the |GCP| > 0.6 threshold needs to be more clearly stated and utilised in the contextualisation of the results. We have edited that section of the manuscript accordingly.

(4b) Likewise, the MR analyses have only a very weak statistical signal (p=.02,.03): 1 this doesn't survive correction for testing two phenotypes (not to mention the implicit tests prioritized by rhog that were discarded based on LCV); 2 the LCV paper proves these tests are susceptible to inflation by genetic correlation; 3 I do not agree that horizontal pleiotropy has been ruled out, a priori it seems almost certain that many heritable traits (BMI, smoking, diet, exercise,.…) will causally effect both glucose and lung function, to some extent, and moreover you do show that AMT has near-significant TWAS effects on both smoking and glucose.

We respectively disagree with the characterisation of the MR data here only providing weak statistical support and respond to each point below:

1) It is true that the inverse-variance weighted estimator with random effects yields nominally significant results in terms of statistical significance, however, this overlooks the robustness, and similarity, of the point estimate using both the MR-Egger and Weighted Median models which have completely different assumptions related to IV validity, as discussed in the manuscript. Furthermore, we would posit that the MR was utilised here as a validation of the relationship observed in the LCV model, and thus, is not our primary evidence for an effect of fasting glucose on lung function. The utility of the MR here is that is does provide a point estimate of the putative effect of the exposure on the outcome, which is not afforded in the LCV framework.

2) We agree with the reviewer that MR estimates may be inflated by genetic correlation, however, our GCP estimates from the LCV model supported the causal relationship, and thus, there is still evidence for a causal effect even if there is some inflation of the MR estimate due to genetic correlation.

3) We also agree the horizontal pleiotropy cannot be ruled out, and indeed, this is not possible statistically or biologically. In spite of that, we perform a number of analyses to try and determine whether there is a confounding effect of horizontal pleiotropy on the MR estimate, as outlined in the manuscript. We believe it is particularly important that we biologically annotated potential outlier instrumental variables as having direct relevance to glycaemic biology, for example, an instrumental variable SNP in the glucokinase gene had a particularly large effect (Supplementary file 1G). We also devote a paragraph to the potential confounding effect of smoking. We did not find any evidence of a genetically causal effect of smoking on fasting glucose any evidence of genetic causality between smoking and fasting glucose via the LCV model, whilst none of the fasting glucose IVs were associated with smoking.

Reviewer #2:In this study, Reay et al. used publicly available GWAS data with regards to lung function and biochemical traits to identify molecular mechanisms capable of improving lung function. There are several comments and questions:1) Reay et al. identified multiple biochemical traits that could be targeted in order to modulate lung function including fasting glucose and fasting insulin levels as well as other glycaemic related pathways and traits. Of these traits they also identify four gene-sets overrepresented with proteins that interact with viral SARS-CoV2 proteins. However as mentioned by Reay et al., previous studies have already found glycaemic control in the form of diabetes to have an effect on both lung function [Klein et al. Diabet Med. 2010] as well as Covid-19 risk and severity [Yang et al., Int J Infect Dis. 2020 94:91-95.]. The results presented here are not groundbreaking on their own.

We respectively disagree with the reviewer on this point, the studies described in our manuscript which previously have supported the relationship between glycaemic biology and/or diabetes with lung function have been *observational* in nature. The key difference in our work is that we leverage genetics to provide novel evidence of potential *causal* relationship between blood glucose and lung function. We achieved this by implementing two approaches: a latent causal variable model and Mendelian Randomisation. These methods and their relationship to causal inference, rather than observational association alone, have been described extensively previously – for instance: PMID: 30374074, PMID: 30002074, PMID: 12689998, PMID: 22607825, PMID: 32249995, and PMID: 29686387. In our manuscript, we also provide an explanation of these methods, as well as their caveats and limitations. We also state that a well-powered, replicated, randomised-control trial would be needed to confirm the causal nature of this relationship. However, our data provides greater justification for an RCT of antihyperglycaemic compounds given the evidence for causality provided by this study. In light of the reviewer’s comments, we have edited the discussion to emphasise the nature of findings relative to previous literature.

2) In addition, the authors have employed several statistical methods using existing datasets and performed a comprehensive analysis. However, all of the approaches are from literature, which limits the novelty of this study.

We would assert in response to the reviewer’s concern that the aim of this study was to synthesise publicly available genomic information to rapidly priortise drug repurposing candidates. As a result, whilst the techniques applied in this study have been developed previously, we believe that the synthesis of causal inference, polygenic scoring, and transcriptomic imputation for drug repurposing in a single integrated study enhances the novelty of the study. It should also be noted that several of our drug repurposing candidates proposed in our study have never previously been identified in the literature, whilst the candidates proposed previously receive greater support resulting from these data.

3) The benefit of using the framework proposed by Reay et al. is that it identifies potential new uses for existing drugs through the biochemical traits they modulate. This however means that the potential discoveries regarding drug repurposing are limited to only those compounds with known biochemical effects. Another limitation is the use of genetics data exclusively and not integrating more layers of information that might identify causal traits for any given disease. Various other approaches have extensively been reported before (e.g. Pushpakom et al. Nat Rev Drug Discov. 2019) which creates the question as to how this methodology can be edited in order to maximize the possible findings.

We agree with the reviewer that a limitation of our work is that we focus on primarily approved compounds and those with characterised molecular effects. Our reasoning for this was to identify drug repurposing candidates that could be utilised most readily, however, this could be expanded in future work. We also acknowledge that other data besides genetics could be integrated for drug repurposing, although this would be the focus on future study. We have added some text to the discussion which acknowledges the reviewer’s comment that other data could be included beyond genotype information.

4) The authors claimed that "The correlation between the expression of genes within each pathway encompassed by the PES and the PES profiles themselves could provide further support for their biological impact". This is true when the expression data of the genes come from the relevant tissues. Here, the authors focus on lung function but performed "association between lung function PES and gene expression using RNA sequencing (RNAseq) on transformed lymphoblastoid cell lines (LCL)". My question is how is LCL relevant for lung function?

We agree with the reviewer that lung tissue would be the ideal tissue to investigate. However, the LCL dataset is a large and easily accessible collection of samples with matched genome and RNA sequencing, and thus, we utilised in this study. We have added a caveat in the respective section of the manuscript, that these relationships should be explored using lung tissue in future.

5) Another question is as to how these findings can be validated either in vivo or in vitro. Figure 5 shows a schematic representation on how treatment could be implemented but it is unclear if any validation experiments have been performed.

We thank the reviewer for this comment. We believe that the ideal validation strategy for this approach would be to test its utility in a clinical trial setting as was outlined in the manuscript. in vitro analysis of polygenic scores remains challenging although approaches like patient derived cell lines would be a useful future avenue.

6) Throughout this study, the authors used three measurements of spirometry phenotypes for lung function. Then, the results and interpretation in this study should be limited to "lung function". However, the authors generalized their observations from "lung function" to "respiratory disease and respiratory infection". This can be misleading (too far-reaching). For example, lung function is often measured by dynamic spirometry which mostly reflects large airway function. However, respiratory disease like COPD is an inflammatory airway disease which affects the small airways in particular (DS Postma, NJEM, ‎2015). Furthermore, "respiratory infection" by bacterial and viral infection such as tuberculosis, influenza and coronaviruses may lead to completely different pathogenesis. It is hard to believe that hyperglycaemia will have causal effect on these respiratory diareses (infections).

While we appreciate the reviewer’s concern that it may not seem logical that such a basic metabolic parameter can have such a broad reaching genetic and epidemiological impact in lung function, however, as we did not hypothesize this the “logic” is not relevant. We discovered this influence using an unbiased (hypothesis-free) data driven approach. As with many things in science, there is not always and immediate explanation of discovery that accords with what is known about a particular phenomenon. While it is possible that not all respiratory disease has an impact on lung function it is likely that lung function will have an impact on lung disease outcome. Lung function is a powerful quantitative trait and a biomarker of lung disease and lung related morbidity/mortality. In other words, regardless of the cause of acute lung disfunction the underlying reserves in lung function will often determine survival and long-term consequences. If these risk factors can be modulated in affected individuals it is likely to have a positive impact on the course of disease.

Given the reviewer’s comments we have modified the text to better explain the clinical significance of the relationship between lung function and lung disease. Furthermore, we have decided to edit the title of this manuscript as follows, which we believe avoids overgeneralising our data: “Genetic association and causal inference converge on hyperglycaemia as a modifiable factor to improve lung function.”

7) For all candidate genes identified by TWAS analysis, they could be further prioritized by checking if they are differentially expressed in the lung between samples with and without impaired lung functions.

This is a sound suggestion by the reviewer, and we have added this to the discussion as a future direction.

8) The authors stated "Probabilistic finemapping of these transcriptome-wide significant regions using a multi-tissue reference panel was then performed to prioritize whether these genes are likely causal at that locus". What is the reasoning behind the prioritization using “multi-tissue reference”? It is known that the majority of cis-eQTLs are shared across tissues, but how this could help to prioritize relevant genes?

Our reasoning for including a multi-tissue panel in our finemapping analyses was to capture the maximum number of genes per locus that have significant *cis*-heritable models. In other words, our TWAS discovery analyses focused on lung and whole blood as likely the most relevant and/or well-powered tissues. The finemapping model then sought to priortise whether our candidate TWAS gene was the most likely causal gene amongst proximally located genes. As a result, we used a multi-tissue panel to include the maximum number of genes in this model when computing the credible set, including genes that did not have a significantly *cis*-heritable in lung or whole blood. The reasoning behind using a multi-tissue panel has been further described elsewhere, such as PMID: 30926970.

Reviewer #3:The manuscript by Reay et al. presents a set of comprehensive analyses of GWAS data to postulate the causal role of hyperglycemia in lung function. The authors perform a series of causal inference analyses on the GWAS data of several blood traits to identify genetically correlated traits that can be explained by a causal role; the authors then seek to identify drug repurposing targets through two complementary analyses, a polygenic risk score restricted to regions within druggable targets and a transcriptome-wide scan linking genetically predicted expression in blood and lung tissue to lung function. Overall, the manuscript leverages recently introduced sophisticate statistical methods and does a thorough job in stress testing the findings. The putative causal role of fasting glucose joint with putative target genes is an important addition to the field. My main comments relate to the robustness of the causal claims.1) The MR analyses assume the blood traits (i.e. fasting glucose) are mediating lung function. Whereas several biological plausible avenues are given in the discussion for this assumption, it can certainly be the case that lung function is mediating fasting glucose (e.g., lung function causing overall body impairment which in turn causes changes in blood measurements). I strongly encourage the authors to perform analyses under this reverse causality assumption. In particular, the bivariate MR method of Pickrell NG 2016 would be relevant here.

We agree with the reviewer that reverse causality is an important consideration when performing causal inference. We believe that our existing data does provide support for the direction of causal effect to primarily operate from fasting glucose to lung function as the sign of the posterior mean GCP is positive in the LCV model. This is likely indicative of the fact that variants exerting an effect on fasting glucose tend to have proportional effects on lung function, but not *vice versa*, as described in the original O’Connor and Price LCV manuscript. We have performed some additional analyses to further support that the direction of assumed effect from exposure to outcome is correct by utilising the MR-Steiger directionality test (PMID: 29149188). The concept underlying this approach is that the estimated phenotypic variance explained by the IV SNPs on the exposure (fasting glucose) and outcome (FEV_1_ or FVC) are compared to establish whether exposure outcome causal effect is correct. We found that this was the case in both instances for FEV_1_ and FVC, with no evidence of reverse causality. This has now been added to the manuscript.

2) As the authors describe in the Discussion section, wrong assumptions in the MR framework can invalidate the findings. The authors do a great job in assaying the impact of pleiotropy on the MR estimates using recently developed methods (LCV, MR-PRESSO etc); however the causal role of smoking is left ambiguous in the causal inference. Clearly smoking has a causal role on lung function, and GWAS of smoking reveals genetic correlates of smoking status (amount). Is there any impact of smoking on blood traits? Is smoking a collider in the causal diagram genetics -> fasting glucose -> lung function? The authors have access to GWAS of smoking and could leverage the MR toolkit to investigate causal effects of smoking on glucose.

We agree with the reviewer that the role of smoking is an important consideration in these results. We believe that our material as described provides support that our results are not an artefact of confounding due to smoking effects, although as ever with genetically informed causal inference, this cannot be definitively ruled out. Firstly, we demonstrated that glucose and smoking were genetically correlated using a GWAS of smoking heaviness. However, using the LCV approach we found no strong evidence of partial genetic causality between smoking and fasting glucose, whilst the GCP estimate was relatively high (GCP = -0.47), the large standard error rendered this point estimate difficult to interpret and statistically non-significant. We believe the most important evidence is that none of our fasting glucose SNPs utilised as IVs were associated with either smoking phenotype at genome-wide or suggestive significance, meaning the MR signal was unlikely to be biased by smoking.

We appreciate the suggestion from the reviewer in regard to whether our estimate may be impacted by collider bias as the lung function GWAS was covaried for smoking status. We performed some additional analyses using a smaller GWAS of FEV_1_ and FVC using the UK biobank cohort only from Ben Neale’s group which was not adjusted for smoking (N = 272,338). We found that the posterior mean GCP estimates and IVW estimates for MR were in the same direction, with no evidence for the role of smoking as a collider variable. This new material has now been added to the manuscript.

3) The identification of drug repurposing tools using the PES score is inconclusive without some replication/validation. The PES is explaining a small proportion of variation in the trait making the interpretation of PES correlations subtle at best; e.g., it is hard to find a biological role for some of the gene-sets that show significance in Table 2. More importantly, it is unclear what is the null expectation of the PES-gene expression correlation analysis; that is, if PES is computed using random pathways (i.e. not specific to druggable pathways) and re-runs the analyses, what are the results? Or, reversely, if the authors perform the same analyses for a randomly chosen complex trait (e.g., height/bmi), what pathways show up in Tables 2/3?

We agree with the reviewer that in future the PES results from our study require replication and validation and have added a statement to this effect in the Discussion. We would also argue that due to the hypothesis free nature of pathway selection (other than having a known drug target), this approach may reveal biological aspects of the phenotype in question that have not previously been considered. The analyses in this study whereby we conservatively control for genome wide PGS we believe suitably addresses the question in regard to the null expectation of the PES ~ lung function correlation. In other words, the class b/2 secretin PES, for instance, was associated with FVC beyond what was attributable to genome wide PGS, suggesting a role for variants in this pathway that is not influenced purely by random subsection of the polygenic signal for this trait amongst these genes. Indeed, the trait relevance of these pathways is exemplified by their selection in that this gene-sets were significantly enriched with trait-associated variation relative to all other genes in the original GWAS. There are likely to be non-druggable pathways to also exert a non-zero effect on lung function, however, given each of the PES were either non-significantly or only modestly correlated with each other, we would expect these signals to be distinct. We have added some text to the discussion that outlines that these concerns warrant further investigation and thank the reviewer for the comment. Furthermore, the pathways identified in this study and utilised for PES are unlikely to be only relevant to lung function given the pleiotropic nature of common variants. Therefore, the association of the same pathways with other traits like BMI or height, in our opinion, does not negate their relevance in this study in the same fashion as a genome-wide significant SNP being shared between two phenotypes does not preclude its importance from either.

[Editors’ note: what follows is the authors’ response to the second round of review.]

Revisions:The joint view of the reviewers and eLife Editors is that several of the reviewers' comments have been addressed adequately. Nevertheless, other reviewers' comments were not resolved by your revision, most importantly Points 1 and 4 of Reviewer 1. As a group we would be more supportive of the revision if the causality claims were to be toned down with appropriate caveats included throughout, and if the statistical results were to be more appropriately presented. The current version, without such changes, is not judged to be ready for publication.We are content that you now acknowledge the LCV publication's recommendation that only tests with |GCP| > 0.6 are considered. However, your revision does not follow this recommendation unswervingly: e.g. "We acknowledge that the posterior mean GCP estimate for the FEV1 does not quite the threshold of > 0.6, and thus, the causal relationship was more rigorous with FVC". Moreover, your revision unnecessarily clouds this important issue: "|GCP| > 0.6 previously postulated to be evidence of a rigorous relationship". We do not support uneven application of an established threshold.It is now acknowledged that some tests were not significant after correcting for multiple tests. Nevertheless, an unwarranted emphasis was sometimes placed on non-significant results, e.g. "However, this still suggested that there was a relationship between the Class B/2 secretin family receptor FVC PES and FVC beyond what is attributable to a genome-wide PGS" and "several gene-sets trended towards surviving correction". Similar problems identified among the MR tests and the adaptive choices for PRS p-value thresholds still need to be addressed. The MR and PES results have relatively weak statistical support and yet this is not reflected by an emphasis placed on them in the Abstract. We are of the view that marginally significant (or null) results can still provide a significant contribution to the field as long as their statistical support is reported appropriately. The manuscript will require changes to the application and interpretation of statistical tests throughout.

We agree with the reviewers that the results of this manuscript need to be carefully presented. As a result, we have made several changes in the manuscript pursuant to this. In particular, we acknowledge the concerns of the reviewers related to the presentation of the LCV, Mendelian randomisation, and PES results. We have addressed each of these sections individually below and detail the edits that have been made to the manuscript.

Pharmagenic enrichment score (PES) results

We have made a number of edits related to the PES materials such that the borderline nature of their statistical association in the Hunter cohort is emphasised to the reader:

Abstract: The language regarding the association between PES and pulmonary phenotypes has been tempered given it does not survive multiple testing correction. Specifically, we have edited the Abstract from “Moreover, we developed polygenic scores for lung function specifically within pathways with known drug targets that were significantly associated with both pulmonary phenotypes and gene expression in independent cohorts to prioritise individuals who may benefit from particular drug repurposing opportunities” to “Moreover, we developed polygenic scores for lung function specifically within pathways with known drug targets and investigated their relationship with both pulmonary phenotypes and gene expression in independent cohorts to prioritise individuals who may benefit from particular drug repurposing opportunities”. We believe that this edited sentence is appropriate in its conservatism as it does not explicitly state the significance of these results, even though we found correlations with mRNA expression that surpassed multiple-testing correction.

In the figure legend for Figure 3, we edited the wording such it states, “The phenotypic association between a polygenic score (PGS) of FVC and an FVC PES which was nominally significant (P < 0.05) but did not survive multiple testing correction after adjustment for genome wide PGS”.

We have removed the mention of gene-sets that trended towards multiple testing correction.

We have removed the sentence “However, this still suggested that there was a relationship between the Class B/2 secretin family receptor FVC PES and FVC beyond what is attributable to a genome-wide PGS”. We believe the description of this result as “nominally significant” but failing to survive correction in the previous line accurately portrays the strength of this result.

We have added some text to further contextualise the strength of these results given that it does not survive multiple testing correction.

The text in the discussion regarding the PES has also been edited to more explicitly state that the Class B/2 secretin family receptor FVC PES association does not survive multiple testing correction – specifically, we changed “The Class B/2 secretin family receptors score for FVC was particularly noteworthy given that it remained significant after an adjustment for genome wide PGS” to “The Class B/2 secretin family receptors score for FVC was noteworthy given that it remained nominally significant after an adjustment for genome wide PGS. However, this did not survive multiple-testing correction, and thus, further replication is needed to confirm this signal”

We also wish to further clarify the selection of *P* value thresholds to construct the PES and the PGS, this issue has also been addressed in a previous publication describing the PES method (PMID: 31964963). Firstly, we use *P* value thresholding in the identification of the candidate PES gene-sets using the GWAS summary statistics, that is, gene-based test statistics are constructed only using *P* values below said threshold. Thereafter, competitive gene-set association is conducted for each druggable pathway at the different thresholds, with the null hypothesis being that the druggable pathway is no more associated with the trait (enriched with association) than all other genes for which gene-based *P* values could be calculated by virtue of having a SNP below the threshold. The concept underlying this is that distinct pathways may be enriched with common variants at differing levels of the polygenic signal, for example, a model including all SNPs (*P* < 1) will identify gene-sets enriched with variation relative to all other genes, whilst a less polygenic model, like a threshold of *P* < 0.05, will capture gene-sets enriched with association relative to genes with at least one SNP mapped to it with a univariate association *P* < 0.05. The selection of the actual thresholds themselves is somewhat arbitrary, however, we justify our selection (*P* < 1, *P* < 0.5, *P* < 0.05, and *P* < 0.005), as this is indicative of a model with all SNPs (*P* < 1), nominally associated SNPs (*P* < 0.05), as well as an order of magnitude higher (*P* < 0.5) or lower (*P* < 0.005) than nominally associated. We believe this balances capturing different elements of the polygenic signal with encompassing enough SNPs to construct scores using variants only within a biological pathway, which may not be realistic at lower *P* thresholds. As described in the text – all the thresholds are considered when applying multiple testing correction via the FDR method, that is, we subject all gene-sets at all thresholds to FDR correction simultaneously, which amounts to around 4000 tests subjected to FDR correction at the same time over all the thresholds. The threshold which is most associated after FDR correction is then used to construct the score for individual-level genotype data. We only ever constructed the PES at one *P* threshold when applying this to individual genotype data, which was the threshold at which the strongest signal was observed in the discovery GWAS, and therefore, we did not need to correct for testing multiple *P* thresholds per PES when we applied this to the Hunter cohort and GEUVADIS datasets, unlike in the GWAS summary statistics analyses where all thresholds were corrected for. We have provided an additional summary of this process to aid in reader understanding, as well as added some text to clarify that only one threshold was tested in the Hunter cohort and GEUVADIS datasets for each PES.

Furthermore, the primary use of genome wide PGS in this study was to act as a covariate in the sensitivity analyses of the PES ~ lung function models in the Hunter cohort and we constructed the PGS at the same thresholds as the PES such that they could be directly compared. However, we also wished to benchmark the variance explained by the PES compared to a genome wide PGS, and, as a result, we constructed and tested genome wide PGS at two additional thresholds that were less polygenic (*P* < 1 x 10^-5^ and *P* < 5 x 10^-8^) to maximise our chances of finding the most parsimonious score with the highest variance explained. The inclusion of more stringent thresholds closer to genome-wide significance when testing PGS is usual practice in the literature. As it turned out, the most parsimonious model was a *P* value threshold of 0.005 and 0.05, for FEV_1_ and FVC, respectively, meaning the addition of these thresholds had no impact on the results. This does not have any implications for multiple testing correction in the PES analyses as we only used the PGS *P* threshold that matched the PES in all instances. The *P* < 1 x 10^-5^ and *P* < 5 x 10^-8^ thresholds would not have been appropriate for the identification of PES pathways as the number of independent SNPs in a single pathway that satisfy these thresholds would likely be insufficient, particularly for small gene-sets. We believe that the above provides an appropriate justification for the *P* value thresholds in this study. In summary, we corrected for all four thresholds in the gene-set discovery process simultaneously using FDR, however, we only constructed a score for each PES using a single threshold, and thus, there was no extra multiple testing burden in the Hunter cohort and GEUVADIS analyses, besides the number of PES tested.

Latent causal variable (LCV) results

We have edited the text in the latent causal variable section of the manuscript to discuss the |GCP| > 0.6 threshold more stringently. We still believe that we have demonstrated strong evidence of partial genetic causality between fasting glucose and lung function given that the GCP estimate for fasting glucose FVC exceeded 0.6 using a fasting glucose GWAS with, and without, BMI covariation. As a result, we believe the evidence for this causal relationship is appropriately reflected in the Abstract where we state “with further evidence of a causal relationship between increased fasting glucose and diminished lung function” Specific edits to this section are outlined below:

We have edited the statement regarding the GCP threshold to more definitively state the utility of this threshold, as described in the original O’Connor and Price manuscript – “We used the recommended threshold for partial genetic causality of |GCP| > 0.6, as this has been demonstrated in simulations to appropriately guard against false positives”.

We have also edited the text related to the LCV results such that the |GCP| > 0.6 threshold is applied in a simplified manner. We also now more explicitly demonstrate that the GCP threshold is not reached partial genetic causality between fasting glucose and FEV_1_. However, given the evidence of partial genetic causality of fasting glucose on FVC, and the FEV_1_ GCP value nearing 0.6 (0.57), we still mention this in text with the caveat that it does not exceed 0.6. We would like to make the point that our application of the GCP threshold here is stringent, although somewhat arbitrary given the GCP estimate is not intended to be interpreted in the same way as something like a *P* value where different thresholds have implicit implications. Nonetheless, we agree with the reviewers that stringency is important in this instance, and thus, we believe we have now more decisively applied the 0.6 threshold in our language in this section.

We have amended Figure 2B such that we denote GCP > 0.6 and GCP < -0.6 with a vertical dotted line as the respective thresholds on the forest plot to make it clearer to the reader that these are our thresholds of interest, rather than zero.

Mendelian randomisation (MR) results

We have also reviewed and amended our language regarding the reporting of the results in the Mendelian randomisation section of the manuscript. We wish to emphasise that given we tested traits for causality that were also genetically correlated, the latent causal variable model is our central test for evidence of a causal relationship because genetic correlation has been shown to potentially bias Mendelian randomisation. As a result, we deploy Mendelian randomisation here as a validation of the relationship observed by LCV as it uses a different set of statistical parameters and assumptions, even though MR may be inflated by genetic correlation. We have edited the text to more explicitly make this point. Given the strength of the LCV model between fasting glucose and FVC, we further would posit the emphasis placed on the relationship of fasting glucose on FVC is still appropriate in the Abstract.

The primary test we utilise for MR is the inverse-variance weighted estimator, as is this is generally considered the most powerful approach, however, this is at the cost of assuming all instrumental variables are valid. We state in text that the IVW estimates were nominally significant (*P* < 0.05), which is true, and we believe is an appropriate description. However, we have added some text to comment that the confidence intervals extend close to zero in both circumstances and that the *P* values are relatively high. The weighted median and MR-Egger methods were then utilised as sensitivity analyses of IVW estimate as they make different assumptions about IV validity. We believe our description of these results is appropriate as we compare them to the IVW method and report their *P* value which was nominally significant in the case of the weighted median and non-significant for MR-Egger. Moreover, we believe that the leave-one-out analyses appropriately demonstrate that the IVW estimate does have some (nominal) outliers, however, these can largely be attributed to what likely constitute true biological effects, as discussed in Supplementary file 1G-L.